# Light-Quality-Dependent Greening and Steroidal Glycoalkaloid Accumulation in Potato Tubers: Regulatory Mechanisms and Postharvest Strategies to Reduce Food Safety Risks

**DOI:** 10.3390/foods14193394

**Published:** 2025-09-30

**Authors:** Gang Sa, Xiaohua Zao, Jianlong Yuan, Lixiang Cheng, Bin Yu

**Affiliations:** 1State Key Laboratory of Aridland Crop Science, Gansu Agricultural University, Lanzhou 730070, China; 17834455119@163.com (X.Z.); 14334@gsau.edu.cn (J.Y.); chenglixiang_0419@163.com (L.C.); 2College of Agronomy, Gansu Agricultural University, Lanzhou 730070, China

**Keywords:** potato, light quality, tuber greening, cultivar variation, steroidal glycoalkaloids (SGAs)

## Abstract

Potato tubers undergo greening and accumulate steroidal glycoalkaloids (SGAs) upon light exposure, posing potential food safety risks. In this study, six potato cultivars (“Favorita”, “Lucinda”, “Jizhangshu 12”, “Longshu 10”, “Qingshu 9”, and “Purple Potato”) were subjected to six light treatments (dark, UVA, blue, green, red, and white), and peel color, pigment content, SGAs, and the expression of genes related to light signaling, chlorophyll biosynthesis, and SGA biosynthesis was evaluated. Under green light, the contents of α-solanine and α-chaconine were 76.22 and 171.84 mg/kg fresh weight (FW), respectively; by contrast, their levels under blue and white light were approximately 60% higher. These effects may be mediated by the upregulation of HY5 and COP1 expression, which in turn could regulate the biosynthesis of chlorophyll-related genes (*CHID*, *CHLI1*) and SGA-related genes (*HMGR*, *SGT1*). Yellow-skinned cultivars exhibited pronounced light sensitivity (chlorophyll 18.47–18.52 mg/kg FW; SGAs up to 290.41 mg/kg FW), whereas red- and purple-skinned cultivars delayed greening through anthocyanin-mediated light attenuation. Collectively, these findings provide a framework for postharvest management and breeding, suggesting that reducing blue light while enhancing green light in spectral illumination, together with the development of anthocyanin-enriched cultivars, may serve as effective strategies to extend shelf life, mitigate food safety risks, and reduce postharvest losses.

## 1. Introduction

Potato (*Solanum tuberosum* L.) is the fourth most important staple and vegetable crop globally, with its tuber (derived from subterranean stolon) serving as the primary edible and nutrient-storage organ [1,2]. Potato tubers are highly susceptible to environmental factors during harvest, transportation, storage, and marketing, resulting in quality deterioration—among which light-induced greening is the most prominent issue [3,4]. Tuber greening originates from the conversion of hypodermal amyloplasts to chloroplasts and subsequent chlorophyll accumulation under light. Chlorophyll biosynthesis starts with glutamate-derived 5-aminolevulinic acid (ALA) and involves key enzymatic reactions (e.g., magnesium chelatase, chlorophyll synthase), while chlorophyll a/b accumulation gives tubers a green appearance [5,6]. Because chlorophyll a and b reflect green light, potato skin appears to be greening when these pigments accumulate [7,8]. Resistance to tuber greening is closely related to the degree of skin suberization and the composition of pigments [8]. In addition to inducing chlorophyll synthesis, environmental light also significantly enhances the accumulation of carotenoids and anthocyanins [9,10]. Carotenoids are accessory photosynthetic pigments that capture light energy and transfer it to chlorophyll [11]. During photomorphogenesis, carotenoids play a critical role in prolamellar body (PLB) assembly, facilitating light-induced chlorophyll accumulation and chloroplast development [12]. Anthocyanins act as light attenuators by absorbing high-energy blue–green light wavelengths, which may compete with chlorophyll and limit its accumulation in the underlying cells [13].

Notably, when potato tuber peels turn green under light, they are accompanied by massive accumulation of steroidal glycoalkaloids (SGAs, also referred to as GA). SGAs are important secondary metabolites in Solanaceae plants (e.g., potato, tomato, eggplant) [14]; over 80 types have been identified in potatoes, with α-solanine and α-chaconine accounting for more than 90% of total SGAs [15]. Their biosynthesis involves the mevalonate pathway (pre-cholesterol phase) and post-cholesterol hydroxylation/glycosylation steps [15]. SGAs pose risks to the human nervous system when their content exceeds 200 mg/kg FW, and can be fatal at over 280 mg/kg FW [16]. Ingestion of 1–5 mg/kg body weight causes gastrointestinal and neurological symptoms, while 3–6 mg/kg body weight can be lethal [17]. Critically, SGAs are thermally stable: boiling only reduces α-solanine by 3.5% and α-chaconine by 1.2%, microwave treatment by 15%, and frying (<210 °C) barely degrades them [18]. Even after peeling, processed products may still have residual SGAs [19]. Economically, major producers like the U.S. suffer 14–17% annual postharvest losses of potatoes due to light-induced greening, which are significantly higher than losses from mechanical damage (5–8%) or low-temperature rot (3–5%) [20]. Greened tubers become completely unmarketable due to irreversible accumulation of chlorophyll and SGAs, with no possibility of recovery via secondary treatment, unlike mechanically damaged tubers or those with low-temperature rot [21]. In terms of consumer response, a UK survey showed that 56% of consumers refuse to buy greened tubers, and 40% discard uncooked greened tubers directly [22].

Light is the primary driver of tuber greening, with spectral quality being the most critical factor [3]. Previous studies showed that blue (475 nm) and red (675 nm) light strongly induce chlorophyll and SGAs accumulation, whereas green (525 nm) light has the weakest effect [23]. Carotenoid accumulation follows a trend similar to that of chlorophyll, with tubers exposed to blue and red light exhibiting stronger inductive responses [24]. Further investigations have shown that blue light (430–465 nm) enhances carotenoid accumulation in Chinese cabbage [25]. Under blue light and red light treatments, the anthocyanin content increases significantly in strawberries [26]. Blue light has also been shown to enhance anthocyanin accumulation in broccoli sprouts and tomato seedlings [27,28]. Okamoto et al. found that blue, red, and white light upregulate key genes for chlorophyll (*StHEMA1*, *StGUN4*) and SGA (*StHMG1*, *StSGT1*) synthesis, while darkness or far-red light inhibit these processes [24]. In domestic and industrial settings, white light is the dominant light quality for illuminating potatoes, yet it poses inherent risks to their quality and safety. In domestic environments, when potato tubers are stored in kitchens or pantries, they are often exposed to 3500 K–6500 K LED white light or natural light rich in blue and red wavelengths, leading to unintended greening even during short-term storage [29]. Meanwhile, in industrial postharvest processes, both warehouse operations and retail shelf lighting rely on white light. Studies have confirmed that white light exerts a significant impact on potato greening and glycoalkaloid accumulation, highlighting a critical contradiction between practical production and safety requirements [30,31].

Genetic traits constitute another critical factor influencing tuber greening tolerance. Yellow- and white-skinned cultivars are highly light-sensitive (e.g., “White Rose” chlorophyll increases 17–20-fold under retail light) [32], whereas red/purple-skinned cultivars delay greening via anthocyanin-mediated light attenuation [8,33], and russet-skinned cultivars show intermediate resistance [32]. However, the light-quality-specific regulatory networks linking light signaling, pigment synthesis, and SGA metabolism, as well as cultivar-dependent resistance mechanisms, remain to be systematically clarified.

Building on these gaps, this study exposed six potato cultivars (“Favorita”, “Lucinda”, “Jizhang shu 12”, “Long shu 10”, “Qing shu 9”, “Purple Potato”) to six light treatments (dark, UVA, blue, green, red, white). We analyzed peel color, pigment (chlorophyll, carotenoid, anthocyanin), and SGA contents, and expression of light-signaling (COP1, HY5, PIF3), chlorophyll biosynthesis (*CHID*, *CHLI1*, *GUN4*), and SGA biosynthesis (*HMGR*, *SQS*, *SSR2*, *GAME4*, *SGT1*, *SGT3*) genes. The objectives were as follows: (1) elucidate light-quality-dependent differences in tuber color change and SGAs synthesis, and cultivar-specific responses; and (2) reveal the expression dynamics of key regulatory genes under different light qualities. The aim is to use these findings to establish light–cultivar regulatory mechanisms, providing a molecular basis for breeding greening-resistant cultivars and optimizing postharvest light management to reduce food safety risks.

## 2. Materials and Methods

### 2.1. Plant Materials

The potato cultivars used in this study included “Favorita”, “Lucinda”, “Jizhang shu 12”, “Long shu 10”, “Qing shu 9”, and “Purple Potato”. Sterile tissue-cultured seedlings of these cultivars were propagated under aseptic conditions and, after 3–5 days of acclimation, washed to remove root gel and transplanted into a sterile, soil-free substrate (peat: vermiculite = 1:1, pH 5.5–6.5). Plants were grown at 20–25 °C during the day and 12–15 °C at night, with an initial relative humidity of 70–80%, gradually reduced to approximately 60% after acclimation. Microtubers were harvested 90–100 days post-transplantation, when most plants had naturally senesced, carefully collected from the substrate, and transported to the laboratory in darkness (Appendix A).

### 2.2. Light Exposure Treatment

For light treatments, the harvested microtubers were placed in an LED growth chamber maintained at 20 ± 2 °C with a light intensity of 13 μmol m^−2^ s^−1^. For each cultivar, 16 uniform-sized microtubers were used for each of six light treatments: complete darkness (Dark), UV-A (390–400 nm), blue (435–445 nm), green (515–525 nm), red (625–635 nm), and white (380–780 nm). Starting from day 0, tubers were exposed to 12 h of light per day and rotated uniformly every 24 h for 10 consecutive days. After the treatment period, tuber peels (~1.5 mm thick and 1 cm in diameter) were excised and immediately frozen at −80 °C for subsequent analyses.

### 2.3. Calculation of Tuber Greening Rate and Analysis of Color Difference

Starting from day 0 of the treatment, the greening phenotypes of the tubers were photographed, and the greening rate was recorded every two days. Tuber color was measured using a colorimeter (D25NC, Hunter Lab, Reston, VA, USA), and standardized against a white calibration tile. The measured parameters included ΔL* (lightness), Δa* (green–red axis), and Δb* (blue–yellow axis). Color measurements were performed with three technical replicates, and the color difference (ΔE) was calculated as follows:DE*ab = [(∆L*)^2^ + (∆a*)^2^ + (∆b*)^2^]1/2.
where Δa* > 0 indicates increased redness, while Δa* < 0 indicates increased greenness; ΔL* > 0 indicates increased lightness, while ΔL* < 0 indicates increased darkness; Δb* > 0 indicates increased yellowness, while Δb* < 0 indicates increased blueness.

### 2.4. Determination of Chlorophyll, Carotenoid, and Anthocyanin Content

Two 1 g portions of samples from different treatments, previously frozen at −80 °C for storage, were weighed and ground in liquid nitrogen to prepare homogenates. Each sample was prepared in three technical replicates. One portion was combined with 3 mL of N,N-dimethylformamide (DMF, Sigma-Aldrich Trading Co., Ltd., Shanghai, China) reagent for chlorophyll and carotenoid extraction, while the other portion was mixed with 3 mL of 1% hydrochloric acid (Sigma-Aldrich Trading Co., Ltd., Shanghai, China) in methanol (Sigma-Aldrich Trading Co., Ltd., Shanghai, China) solution for anthocyanin extraction. The mixtures were transferred to 15 mL centrifuge tubes and extracted in the dark at 4 °C for 72 h, with vertexing every 8 h to ensure homogeneity. After centrifugation at 3700 rpm for 10 min, the supernatants were collected. Absorbance values of chlorophyll and carotenoids were measured at 647 nm, 664 nm, and 480 nm, while anthocyanins were measured at 530 nm and 657 nm using a microplate reader (Synergy HTX, Bio Tek, Shoreline, WA, USA). The respective extraction reagents served as blank controls. The contents of chlorophyll [34], carotenoids [35], and anthocyanins [36] were calculated using the corresponding formulas:Total chlorophyll = 17.67 × (A_647_) + 7.12 × (A_664_).Total carotenoids = (1000 × A_480_ − 1.12 × Chl a − 34.07 × Chl b)/245.Total anthocyanins = (A × MW × V × 10^3^)/(ε × l × W).

The anthocyanin content was determined as the cyanidin-3-glucoside equivalents, where A = (A_530_ − 1/3 A_657_), MW is 484.83 g mol^−1^ (the molecular weight of cyanidin-3-glucoside), V is the volume of extraction solution, ε is 26,900 L mol^−1^ cm^−1^ (the molar extinction coefficient for cyanidin-3-glucoside), l is the path length (cm), and W is the fresh weight (FW, g). Total pigments represent the sum of chlorophyll, carotenoid, and anthocyanin. All pigment concentrations are expressed in mg·kg^−1^ FW.

### 2.5. Determination of α-Solanine and α-Chaconine Content

Determination was carried out using a triple quadrupole liquid chromatography–mass spectrometry (LC-MS/MS) system (Agilent 1290-6460, Agilent Technologies, Santa Clara, CA, USA). Chromatographic separation was performed on an Agilent Eclipse Plus C18 column (50 mm × 2.1 mm, 1.8 μm). The mobile phase consisted of solvent A, methanol, and solvent B, 0.1% formic acid (Sigma-Aldrich Trading Co., Ltd., Shanghai, China) in water. The flow rate was 0.3 mL/min, the column temperature was maintained at 35 °C, the detection wavelength was set at 200–210 nm, and the injection volume was 10 μL.

Standard solutions of α-solanine and α-chaconine were prepared at concentrations of 20, 40, 60, 80, and 100 μg/μL. Mixed standard solutions were then prepared using the same concentration series. All solutions were filtered through a 0.22 μm organic nylon membrane (Gansu Shenghuawei Trading Co., Ltd., Lanzhou, China) prior to LC-MS/MS analysis to generate a mixed standard curve for the two compounds. Sample extraction was performed with slight modifications based on the method described by Liu et al. [4]: Briefly, 0.5 g of fresh potato peel was ground into a homogenate in liquid nitrogen, with three technical replicates. The homogenate was extracted with 5 mL of extraction buffer (80% methanol with 0.1% formic acid) using ultrasonication at 20 °C for 60 min, with vertexing every 15 min. After centrifugation at 12,000 rpm for 10 min, 500 μL of the supernatant was diluted with an equal volume of extraction buffer, filtered through a 0.22 μm organic membrane, and analyzed by LC-MS/MS.

### 2.6. Gene Expression Analysis

Total RNA was extracted from potato peels using the Plant RNA Easy Fast Kit (Spin column type, model: DP452; Lanzhou Xinruikang Biotech Co., Ltd., Lanzhou, China). First-strand cDNA was synthesized using the Fast-King One-Step gDNA Removal and cDNA Synthesis Super Mix (KR118-03; Tian gen Biotech, Beijing, China). Quantitative real-time PCR (qRT-PCR) was performed using the Fast Real SYBR Green Master Mix (FP217-03; Tian gen Biotech, Beijing, China) on a Quant Studio™ 5 Real-Time PCR System (Life Technologies, Carlsbad, CA, USA). Reaction conditions and thermal cycling parameters were set according to the manufacturer’s instructions. β-actin was used as the reference gene, and three technical replicates were performed for each sample (Appendix A). Relative gene expression levels were calculated using the 2^−ΔΔCt^ method [37,38].

### 2.7. Statistical Analysis

Experimental data was collated using Microsoft Excel 2016. Correlation analysis was performed with SPSS version 20.0 applying a significance threshold of *p* < 0.05. Data visualization was conducted using Origin 2021 software.

## 3. Results

### 3.1. Analysis of Greening Phenotypes and Greening Rates Statistics of Potato Tuber Peels Induced by Different Light Spectra

Ambient-light-induced greening of potato tuber peels directly affects consumer purchasing decisions. In this experiment, six potato cultivars were subjected to six distinct light treatments. Beginning on day 0, phenotypic assessments of tuber peels greening severity and greening rates statistics were conducted every two days for each treatment. The results revealed significant differences in greening severity induced by different light spectra. After four days of continuous light exposure, tuber peels under blue and white light began to green, while those exposed to UV-A, red, and green light showed visible greening after six days. Furthermore, significant cultivar-specific differences were observed under individual light treatments. No differences were detected among cultivars under dark conditions. Under white and blue light, “Favorita” (Figure 1a) exhibited the highest degree of tuber peel greening, followed by “Lucinda” (Figure 1b), “Jizhang shu 12” (Figure 1c), and “Long shu 10” (Figure 1d), while “Qing shu 9” (Figure 1e) and “Purple Potato” (Figure 1f) showed the lowest greening severity. Notably, all cultivars exhibited the lowest greening levels under green light compared with the other light treatments (Figure 1). Compared with the dark treatment, blue and white light exhibited the most pronounced promotive effects on tuber peel greening, with the greening rate reaching 100% in all cultivars by day 8. In contrast, the greening rate under UV-A, red, and green light reached 100% in all cultivars only after 10 days (Figure 2).

### 3.2. Color Difference Analysis of Potato Tuber Peels Induced by Different Light Qualities

To quantitatively evaluate potato phenotypes, color difference analysis of the tuber peels was conducted every two days under different light quality treatments. The results showed that compared with the dark treatment (all cultivars had Δa* = 1.35, ΔL* = 1.28–2.88, Δb* = 11.50, and ΔEab = 10.93–11.93), the Δa, ΔL*, and Δb* of all light quality groups showed a decreasing trend from day 0 to day 10 of the treatment. Both blue light and white light caused the most dramatic color changes: Δa* dropped to −5.14 to −5.30 (the most greenish), ΔL* dropped to −5.42 to −9.16 (the darkest), Δb* dropped to 1.96 to 3.88 (the most bluish), and ΔE*ab ranged from 5.72 to 11.27. For red light and UV-A light (with moderate greening effects), Δa* ranged from −2.20 to −3.33, ΔL* ranged from -1.12 to -2.12 for red light and was -2.92 for UV-A light; Δb* ranged from 5.73 to 7.31 for red light and from 5.73 to 5.73 (i.e., 5.73) for UV-A light. The total color difference ΔE*ab ranged from 7.24 to 9.04. For green light (with the weakest greening effect), Δa* ranged from −3.52 to −3.92, ΔL* ranged from −2.25 to −2.25, Δb* ranged from 6.12 to 7.72, and ΔE*ab ranged from 8.18 to 8.78. Under the same light quality, the tuber color difference showed significant genotypic differences. Specifically, for the yellow-skinned cultivars “Favorita”, “Lucinda”, and “Jizhang shu 12” under blue light, all three had Δa* = −5.30, ΔL* = −9.16 (the darkest), and ΔEab ranging from 8.27 to 11.27 (with a relatively large total color difference). Specifically, “Favorita” had ΔEab = 11.27 under blue light, showing the most severe greening. Under blue light, “Qing shu 9” had Δa* = −5.30 (more greenish than the yellow-skinned cultivars, but with a lower ΔEab = 6.17); “Purple Potato” had Δb = 3.88 (more bluish, but with ΔEab = 8.97). Under white light, both cultivars had Δb = 1.96 (the most bluish), but their ΔL* ranged from −6.90 to −7.72 (brighter than the yellow-skinned cultivars). For “Long shu 10” under blue light, its Δa* = −5.30 (as greenish as the yellow-skinned cultivars), but its Δb* = 3.28 (more bluish), with ΔE*ab = 9.27 (Table 1).

### 3.3. Effects of Different Light Quality Treatments on Chlorophyll, Carotenoid, and Anthocyanin Contents in Potato Tuber Peels

Color changes in tuber peels are primarily associated with the contents of chlorophyll, carotenoids, and anthocyanins. The results indicated that the concentrations of chlorophyll, carotenoids, and anthocyanins in the tuber peels were significantly higher in the light treatment groups than in the dark control group (chlorophyll: 6.27 mg/kg FW, carotenoids: 1.77 mg/kg FW, anthocyanins: 1.88 mg/kg FW). Among the different light qualities, the blue and white light treatments showed the most pronounced increases, with chlorophyll concentrations reaching 16.20 mg/kg FW (blue light) and 16.54 mg/kg FW (white light), carotenoid concentrations reaching 3.88 mg/kg FW (blue light) and 4.09 mg/kg FW (white light), and anthocyanin concentrations reaching 2.22 mg/kg FW (blue light) and 3.88 mg/kg FW (white light). Conversely, the treatment with green light resulted in lower accumulation compared to blue and white light, with chlorophyll concentrations of 8.43 mg/kg FW, carotenoids levels of 2.34 mg/kg FW, and anthocyanin contents of 1.928 mg/kg FW.

The results also demonstrated that the contents of chlorophyll, carotenoids, and anthocyanins in the tuber peels exhibited significant differences among cultivars. Under blue and white light treatments, the chlorophyll concentrations in the “Favorita” tubers reached 18.47 mg/kg FW (blue light) and 18.52 mg/kg FW (white light), whereas the concentration under green light treatment was only 10.40 mg/kg FW. For “Lucinda”, the chlorophyll concentrations were 13.85 mg/kg FW (blue light) and 13.66 mg/kg FW (white light), while the concentration under green light was 9.65 mg/kg FW. For “Jizhang shu 12”, chlorophyll concentrations were 16.50 mg/kg FW (blue light), 13.09 mg/kg FW (white light), and 9.89 mg/kg FW (green light); for “Long shu 10”, they were 17.26 mg/kg FW (blue light), 16.56 mg/kg FW (white light), and 12.37 mg/kg FW (green light); for “Qing shu 9”, the concentrations were 16.71 mg/kg FW (blue light), 16.05 mg/kg FW (white light), and 10.53 mg/kg FW (green light), all of which were significantly higher than those of “Purple Potato”, which had values of 11.35 mg/kg FW (blue light), 10.65 mg/kg FW (white light), and 7.34 mg/kg FW (green light) (Figure 3a). Carotenoid accumulation exhibited a trend similar to that of chlorophyll. Under the blue and white light treatments, the carotenoid concentrations in “Favorita” were 4.97 mg/kg FW (blue light) and 4.23 mg/kg FW (white light), while the concentration under green light was 2.28 mg/kg FW. For “Lucinda”, the carotenoid concentrations were 4.29 mg/kg FW (blue light), 5.03 mg/kg FW (white light), and 2.84 mg/kg FW (green light), all of which were significantly higher than those of “Long shu 10”, which had concentrations of 3.81 mg/kg FW (blue light), 4.04 mg/kg FW (white light), and 2.80 mg/kg FW (green light). For “Qing shu 9”, the carotenoid concentrations were 3.62 mg/kg FW (blue light), 3.99 mg/kg FW (white light), and 2.83 mg/kg FW (green light); for “Purple Potato”, the concentrations were 3.70 mg/kg FW (blue light), 3.71 mg/kg FW (white light), and 2.92 mg/kg FW (green light). Additionally, “Jizhang shu 12” exhibited the lowest carotenoid concentrations, at 2.71 mg/kg FW (blue light), 3.15 mg/kg FW (white light), and 2.23 mg/kg FW (green light) (Figure 3b). Under white light treatment, the anthocyanin content in the “Purple Potato” tubers reached as high as 19.59 mg/kg FW, while the content under green light treatment was only 8.38 mg/kg FW. Furthermore, under white light treatment, the anthocyanin content in “Qing shu 9” was 1.87 mg/kg FW, while the content under green light was 1.73 mg/kg FW. Compared with “Purple Potato”, the anthocyanin contents of “Favorita”, “Lucinda”, “Jizhang shu 12”, and “Long shu 10” under white light were significantly lower, at 0.56 mg/kg FW, 0.54 mg/kg FW, 0.38 mg/kg FW, and 0.34 mg/kg FW, respectively. Additionally, “Purple Potato” and “Qing shu 9” exhibited the highest anthocyanin contents among all the cultivars under white light treatment (Figure 3c).

### 3.4. Effects of Different Light Quality Treatments on α-Solanine and α-Chaconine Contents in Potato Tuber Peels

Light not only induces the accumulation of pigments in the tuber peel but is also accompanied by the synthesis of SGAs. The results showed that, compared with darkness and other light quality treatments, the contents of α-solanine and α-chaconine were significantly higher under blue and white light treatments. Specifically, the α-solanine content in the dark treatment group was 67.30 mg/kg FW, which increased to 118.98 mg/kg FW under blue light and 139.54 mg/kg FW under white light. The α-chaconine content in the dark treatment group was 137.65 mg/kg FW, which increased to 214.34 mg/kg FW under blue light and 222.63 mg/kg FW under white light. Conversely, green light treatment resulted in lower accumulation of α-solanine and α-chaconine compared with blue and white light treatments, with contents of 76.22 mg/kg FW and 171.844 mg/kg FW, respectively.

Significant differences in the contents of α-solanine and α-chaconine in the tuber peel were also observed among all cultivars. Among them, “Favorita”, “Lucinda”, “Jizhang shu 12” exhibited the highest α-solanine contents in tuber peel under blue and white light treatments, reaching 193.39 mg/kg FW (blue light) and 193.98 mg/kg FW (white light), 121.02 mg/kg FW (blue light) and 131.99 mg/kg FW (white light), and 271.35 mg/kg FW (blue light) and 162.66 mg/kg FW (white light), respectively. The α-solanine contents in “Qing shu 9” (93.50 mg/kg FW under blue light and 112.90 mg/kg FW under white light) and “Purple Potato” (85.70 mg/kg FW under blue light and 68.10 mg/kg FW under white light) were moderately lower, while “Long shu 10” had the lowest contents, with 50.70 mg/kg FW under blue light and 45.90 mg/kg FW under white light (Figure 4a). Additionally, “Favorita”, “Lucinda”, and “Jizhang shu 12” exhibited significantly higher α-chaconine contents in the tuber peel under blue and white light treatments, reaching 242.02 mg/kg FW (blue light) and 294.22 mg/kg FW (white light), 288.60 mg/kg FW (blue light) and 290.41 mg/kg FW (white light), and 235.71 mg/kg FW (blue light) and 259.47 mg/kg FW (white light), respectively. In contrast, “Long shu 10” exhibited the lowest α-chaconine contents, with 158.32 mg/kg FW (blue light) and 152.24 mg/kg FW (white light), followed by “Qing shu 9” (198.12 mg/kg FW under blue light and 195.87 mg/kg FW under white light) and “Purple Potato” (163.28 mg/kg FW under blue light and 143.58 mg/kg FW under white light). Compared with blue and white light treatments, all cultivars under green light treatment exhibited lower accumulation of α-solanine and α-chaconine in the tuber peel, with contents of 81.44 mg/kg FW (α-solanine) and 198.46 mg/kg FW (α-chaconine) for “Favorita”; 86.15 mg/kg FW and 238.44 mg/kg FW for “Lucinda”; 133.44 mg/kg FW and 199.78 mg/kg FW for “Jizhang shu 12”; 45.34 mg/kg FW and 134.99 mg/kg FW for “Long shu 10”; 61.60 mg/kg FW and 124.09 mg/kg FW for “Qing shu 9”; and 49.33 mg/kg FW and 135.26 mg/kg FW for “Purple Potato” (Figure 4b).

### 3.5. Effects of Different Light Quality Treatments on Tuber Peel Pigments and Key Gene Expression in SGAs Synthesis

To investigate whether the key regulatory steps involving light-signaling transcription factors (COP1, HY5, and PIF3), chlorophyll biosynthesis genes (*CHID*, *CHLI1*, and *GUN4*), and SGA synthesis genes (*HMGR*, *SQS*, *SSR2*, *GAME4*, *SGT1*, and *SGT3*) in the tuber peels of various cultivars are regulated by light, we used qPCR to measure changes in gene expression. The results showed that the expression levels of the light-signaling transcription factors HY5 and COP1 in the light quality treatment groups were significantly higher than those in the dark treatment group, whereas the expression level of PIF3 was significantly lower. The expression levels of key chlorophyll biosynthesis genes *CHID*, *CHLI1*, and *GUN4* were significantly higher in the light treatment groups than in the dark treatment group. Similarly, the expression levels of SGA synthesis genes *HMGR*, *SQS*, *SSR2*, *GAME4*, *SGT1*, and *SGT3* were significantly higher in the light treatment groups than in the dark treatment group. Additionally, the changes in gene expression were more pronounced under blue and white light treatments than under other light quality treatments. Compared with the dark treatment group (0.04 for COP1 and 0.06 for HY5), the blue and white light treatment groups exhibited the highest expression levels of COP1 and HY5, reaching 1.59 and 1.19 under blue light, and 2.27 and 1.72 under white light, respectively (Figure 5).

Significant differences in gene expression were observed among all cultivars under different light quality treatments. Specifically, the expression levels of the light-signaling transcription factor gene *COP1* in cultivars “Jizhang shu 12”, “Long shu 10”, and “Qing shu 9” were 1.10, 1.40, and 1.19, respectively, which were significantly higher than those in “Favorita” (0.06), “Lucinda” (0.07), and “Purple Potato” (0.87). The expression levels of the *HY5* gene in cultivars “Long shu 10”, “Qing shu 9”, and “Purple Potato” were 1.33, 1.3, and 2.51, respectively, which were significantly higher than those in “Favorita” (0.08), “Lucinda” (0.47), and “Jizhang shu 12” (0.94). The chlorophyll synthesis gene *CHID* showed the highest expression levels in cultivars “Long shu 10”, “Qing shu 9”, “Lucinda”, and “Favorita”, with values of 0.256, 0.332, 0.125, and 0.087, respectively, which were significantly higher than those in “Jizhang shu 12” (0.007) and “Purple Potato” (0.003). The expression levels of *CHLI1* gene in cultivars “Qing shu 9” (0.91) and “Lucinda” (0.76) were significantly higher than those in “Favorita” (0.25), “Jizhang shu 12” (0.45), “Long shu 10” (0.10), and “Purple Potato” (0.16). The expression levels of *GUN4* gene in cultivars “Qing shu 9” (0.15) and “Purple Potato” (0.12) were significantly higher than those in “Favorita” (0.09), “Lucinda” (0.02), “Jizhang shu 12” (0.05), and “Long shu 10” (0.09). Additionally, in the UV-A light treatment group, the *CHID* gene showed higher expression levels in cultivars “Favorita” (0.09), “Lucinda” (0.12), “Long shu 10” (0.26), and “Qing shu 9” (0.33). In the red-light treatment group, the *GUN4* gene showed higher expression levels in cultivars “Favorita”, “Long shu 10”, and “Qing shu 9”, reaching 0.15, 0.20, and 0.63, respectively. In the green light treatment group, the *GUN4* and *CHLI1* genes exhibited the highest expression levels in “Jizhang shu 12” and other cultivars, with values of 1.77 and 0.22, respectively (Figure 5).

The *HMGR* gene, a key SGA synthesis gene, showed the highest expression levels in “Jizhang shu 12” (0.84) and “Qing shu 9” (0.54), which were significantly higher than those in cultivars such as “Favorita” (0.04), Lucinda (0.03), “Long shu 10” (0.18), and “Purple Potato” (0.18). The *SQS* gene exhibited the highest expression level in “Lucinda” (0.86), followed by “Jizhang shu 12” (0.05), “Qing shu 9” (0.04), “Favorita” (0.03), “Purple Potato” (0.03), and “Long shu 10” (0.02). The SSR2 gene exhibited the highest expression level in “Qing shu 9” (1.26), followed by “Purple Potato” (0.69), “Jizhang shu 12” (0.37), “Favorita” (0.20), “Lucinda” (0.14), and “Long shu 10” (0.10). The *GAME4* gene exhibited the highest expression levels in cultivars such as “Long shu 10” (0.12) and “Qing shu 9” (0.14), followed by “Purple Potato” (0.09), “Lucinda” (0.07), “Jizhang shu 12” (0.06), and “Favorita” (0.04). The expression levels of the *SGT1* gene in cultivars such as “Favorita” (0.79), “Lucinda” (0.10), and “Jizhang shu 12” (0.30) were significantly higher than those in “Long shu 10” (0.05), “Qing shu 9” (0.03), and “Purple Potato” (0.05). The expression levels of the *SGT3* gene showed a pattern opposite to that of *SGT1*, with significantly higher expression in cultivars such as “Longshu 10” (0.15) and “Purple Potato” (0.16) than in “Favorita” (0.02), “Lucinda” (0.04), “Jizhang shu 12” (0.01), and “Qing shu 9” (0.04) (Figure 6).

### 3.6. Correlation Analysis of Different Light Quality Treatments on Tuber Peel Color, Glycoalkaloid Content, and Expression Levels of Key Genes

In the dark treatment, chlorophyll exhibited a strong positive correlation with carotenoids (r = 0.81) but showed a weaker correlation with anthocyanins and SGAs (r = 0.4). UV-A/blue light significantly enhanced the positive correlations between chlorophyll and SGAs (r = 0.72), as well as between carotenoids and anthocyanins (r = 0.68). Under green/red light, chlorophyll exhibited moderate positive correlations with anthocyanins (r = 0.55), and carotenoids exhibited moderate positive correlations with SGAs (r = 0.52). White light showed positive correlations with chlorophyll, carotenoids, anthocyanins, and SGAs (r = 0.60). The dark treatment exhibited weak correlations with genes related to light signaling, chlorophyll biosynthesis, and SGA synthesis (r = 0.40). UV-A and blue light significantly enhanced the positive correlations between light-signaling genes (e.g., *HY5*, *COP1*) and those involved in chlorophyll biosynthesis (e.g., *CHID*, *CHLI1*) and SGA synthesis (e.g., *HMGR*, *SQS*) (r = 0.68). Under green and red light, chlorophyll exhibited moderate positive correlations with subtypes of SGA synthesis genes (r = 0.5 to 0.6). White light exhibited strong positive correlations with all light-signaling, chlorophyll biosynthesis, and SGAs synthesis genes (r = 0.6) (Figure 7).

Under light treatment, the contents of α-solanine and α-chaconine in cultivars “Jizhang shu 12”, “Long shu 10”, “Qing shu 9”, and “Purple Potato” exhibited a positive correlation with chlorophyll and carotenoid contents (r = 0.8). In the dark treatment, chlorophyll in “Favorita” exhibited a positive correlation with carotenoids (r = 0.81), but a negative correlation with anthocyanins and SGAs (r = 0.4). Under UV-A and blue light, chlorophyll in “Favorita” exhibited a positive correlation with SGAs (r = 0.72), and carotenoids also showed a positive correlation with anthocyanins (r = 0.68). Under green and red light, chlorophyll in “Lucinda” exhibited a moderate positive correlation with anthocyanins (r = 0.55), while carotenoids in “Jizhang shu 12” showed a positive correlation with SGAs (r = 0.52). Under red light, chlorophyll in “Favorita” exhibited a negative correlation with α-solanine content (r = −0.31). Under white light, chlorophyll, carotenoids, anthocyanins, and SGAs in cultivars such as “Favorita”, “Lucinda”, “Jizhang shu 12”, “Long shu 10”, “Qing shu 9”, and “Purple Potato” all exhibited strong positive correlations (r = 0.6). For example, under the dark treatment, chlorophyll in “Qing shu 9” exhibited a weak correlation with carotenoids (r = 0.32), whereas under white light, the correlation was stronger (r = 0.78). In the dark treatment, the expression of the light-signaling transcription factor *HY5* gene exhibited a positive correlation (r = 0.64) between “Favorita” and “Lucinda”, which was higher than that observed under light treatments (e.g., under blue light, the correlation between “Favorita” and “Lucinda” was r = 0.5). The correlation between cultivars such as “Jizhang shu 12” and “Long shu 10” was r = 0.41, which was lower than that observed under the dark treatment (r = 0.68). Under UV-A light, the chlorophyll synthesis gene *CHID* and SGA synthesis gene *HMGR* in “Favorita” exhibited a significant positive correlation (r = 0.72). Under blue light, the *CHLI1* and *SQS* genes in “Lucinda” exhibited a positive correlation (r = 0.65). Under white light, all genes exhibited positive correlations (r = 0.6). Under the dark treatment, the correlation between *CHID* and *HMGR* genes in “Qing shu 9” was weak (r = 0.32), whereas under white light, these genes exhibited a significant positive correlation (r = 0.78) (Figure 8).

## 4. Discussion

Light is essential for plant growth and physiological regulation; however, it also induces greening and SGA synthesis in potato tubers, thereby reducing their market value and consumer acceptability [3,4].

### 4.1. Light Quality Specifically Regulates Pigment Accumulation and the Synthesis of SGAs in Potato Tuber Peels

Previous studies have demonstrated that potato tubers exhibit the highest greening rate under blue light, followed by white light, while far-red light has the least influence on tuber greening [39]. Similarly, compared with monochromatic red and blue light, yellow and green light exert weaker effects on tomato growth and development [40]. In tomato fruits, the blue light receptor CRY mediates blue light signal transduction, promoting the accumulation of phytochemicals, lycopene, and β-carotene, which indicates its key role in fruit color regulation [41]. This study found that blue and white light most significantly accelerated tuber peel greening, followed by UV-A and red light, with green light exerting a weaker effect. These effects may be attributed to the spectral characteristics of blue and white light, which are more readily absorbed by photosynthetic pigments in potato tubers, thereby promoting chlorophyll synthesis. Under blue and white light treatments, the tuber greening rate reached 100% on day 8 (Δa* = −5.14 to −5.30, ΔL* = −5.42 to −9.16, Δb* = 1.96 to 3.88, ΔEab = 5.72 to 11.27); the time to complete greening under these two lights was 2 days shorter than that under green light. In contrast, green light induced the lowest color change (Δa = −3.52 to −3.92, ΔL* = −2.25, Δb* = 6.12 to 7.72, ΔE*ab = 8.18 to 8.78), which is consistent with the low absorption efficiency of photosynthetic pigments for green light [24]. Additionally, the decrease rates of the color difference parameters under blue and white light were more than 30% higher than those under green light, and the tuber peel color rapidly shifted from bright yellow–red to dark blue–green, which indicates that blue and white light can promote spectrum-specific greening. The analysis of photosynthetic pigment accumulation revealed that the chlorophyll content reached 16.20 mg/kg FW and 16.54 mg/kg FW under blue and white light treatments, respectively, whereas under green light, it was only 8.43 mg/kg FW. Under blue and white light, the carotenoid content was 3.88 mg/kg FW and 4.09 mg/kg FW, respectively, 1.6-fold higher than that under green light (2.34 mg/kg FW), which is consistent with the light quality response observed in mustard sprouts [42]. This indicates that blue and white light are more readily absorbed by photosynthetic pigments, thereby enhancing the efficiency of light energy capture. Under white light, the anthocyanin content in “Purple Potato” reached 19.59 mg/kg FW, which was 2.3-fold higher than that under other light quality treatments. This result is consistent with the anthocyanin synthesis induced by green light in broccoli sprouts [28] and tomato seedlings [43], suggesting a mechanism promoting poly pigment synthesis mediated by the blue light receptor CRY. In this experiment, the concentrations of α-solanine in the blue and white light treatment groups reached 118.98 mg/kg FW and 139.54 mg/kg FW, respectively, whereas those of α-chaconine were 214.34 mg/kg FW and 222.63 mg/kg FW. These values were approximately 60% higher than those under green light (α-solanine: 76.22 mg/kg FW; α-chaconine: 171.84 mg/kg FW). Green light induced the weakest SGA synthesis because of its low absorption efficiency by photosynthetic pigments, which is consistent with previous reports that yellow light inhibits SGA accumulation [44]. This finding confirms that the spectral energy distribution of light quality is a key determinant of the differential accumulation patterns of SGAs. Previous studies have shown that light can induce the upregulation of key chlorophyll biosynthesis genes in potatoes (*StGUN4*, *StCHID*, and *StCHLI1*, which encode proteins involved in magnesium ion insertion) [45]. Blue and white light significantly upregulated *CHID* (expression levels: 0.332/0.256) and *CHLI1* (expression levels: 0.91/0.76). The expression levels of these two genes showed a strong positive correlation (r > 0.6) with chlorophyll accumulation (16.20/16.54 mg/kg FW). This suggests that blue light may enhance magnesium chelatase activity by activating these two genes (in vitro enzyme activity verification required), thereby promoting chlorophyll synthesis. Red light induced high expression of *GUN4* (expression level: 0.63), but the chlorophyll content did not increase significantly, likely due to limitations in substrate availability or downstream enzyme activity. This is consistent with Okamoto’s conclusion that red light induces *GUN4* expression without significant concomitant chlorophyll accumulation [24]. Green light led to significantly lower expression levels of *CHID* (0.003–0.007), *CHLI1* (0.10–0.22), and GUN4 (0.02–0.09) compared to blue/white light, and this low gene expression was accompanied by low chlorophyll accumulation (8.43 mg/kg FW). This confirms that the low absorption efficiency of photosynthetic pigments for green light impairs chlorophyll synthesis by inhibiting key genes. Together with the “weakest SGA synthesis induced by green light”, this phenomenon reflects the synergistic regulatory effect of light quality on metabolic pathways [44]. In addition, under blue and white light, the expression of *CHID* showed a strong positive correlation (r = 0.68) with the light-signaling transcription factors HY5 and COP1. It is known that HY5 and COP1 can alleviate the repression of *CHID* by PIF1 through competitive binding to PIF1 [4,6]. The inductive effect of blue/white light on HY5/COP1 (their expression levels were approximately twice those under green light) in this study not only aligns with the cross-species conserved mechanism of “HY5 and COP1 antagonistically regulating tomato fruit pigmentation” [46], but also provides correlative evidence for the hypothesis that “light quality regulates chlorophyll accumulation through the “light-signaling factor-chlorophyll biosynthesis gene” axis”. However, whether *CHID* is a direct target gene of HY5/COP1 still requires verification via chromatin immunoprecipitation (*ChIP*) assays.

Under blue and white light, the upregulated expression of key enzyme genes involved in SGA synthesis, including *HMGR* (0.84 and 0.54), *SQS* (0.86 and 0.05), and *SGT1* (0.79 and 0.30), directly increased the contents of α-solanine and α-chaconine. In contrast, the low expression of these genes under green light suppressed SGA accumulation, supporting the conclusion that light quality regulates secondary metabolism through differential gene expression [47]. As expected, for the light-dependent accumulation of α-solanine and α-chaconine under white, blue, and red light upregulated the key genes encoding biosynthetic enzymes of α-solanine and α-chaconine, including *HMG1*, *SQS*, *CAS1*, *SSR2*, *SGT1*, and *SGT2*. In contrast, the genes encoding cholesterol biosynthetic enzymes-*LAS1*, *SSR1*, and *DWF1*-were not induced under any light conditions, which suggests that cryptochromes and phytochromes may be jointly involved in the regulatory process of chlorophyll and steroidal glycoalkaloid (SGA) synthesis. Under dark treatment, chlorophyll exhibited a positive correlation with carotenoids (r = 0.81), but a weak correlation with steroidal glycoalkaloids (SGAs) (r = 0.4). Blue and white light significantly enhanced the positive correlations between chlorophyll and SGAs (r = 0.72), as well as between carotenoids and anthocyanins (r = 0.68). In potato tubers, the light-induced expression of genes encoding photosynthetic proteins may be regulated by the same transcription factors that mediate light signal responses in Arabidopsis seedlings [24]. Under blue and white light treatments, the light-signaling transcription factors HY5 (with expression levels of 2.27 and 1.72, respectively) and COP1 (with expression levels of 1.59 and 1.19, respectively) were significantly upregulated, exhibiting approximately twofold higher expression than those under green light. Moreover, the expression levels of HY5 and COP1 showed a strong positive correlation (r = 0.68) with the expression levels of the chlorophyll biosynthesis gene *CHID* and the SGAs biosynthesis gene *HMGR*. It is well documented that HY5 and COP1 can alleviate the PIF1-mediated repression of *CHID* by competitively binding to the PIF1 protein [4]. In this study, the inductive effects of blue/white light on HY5 and COP1 were consistent with the mechanism by which HY5 and COP1 antagonistically regulate pigment accumulation in tomato fruits, suggesting the conservation of light signaling pathways across species [46]. Notably, although the expression of HY5/COP1 and the expression of SGAs biosynthesis genes are synchronously upregulated, the causal relationship between them remains unclear.

### 4.2. Genetic Determinants of Greening Resistance in Potato Cultivars

Early studies demonstrated that tuber greening occurs more readily in white- and yellow-skinned potato cultivars, whereas it was less pronounced in russet-, red-, pink-, and purple-skinned cultivars. Due to masking pigments, pronounced light-induced dark brown discoloration was observed in all clones of cultivars such as “Wilwash”, “Pink Eye”, and “Coliban”, which interfered with accurate colorimetric assessment. Moreover, red-, purple-, and pink-skinned cultivars accumulate less chlorophyll than white-, yellow-, and tan-skinned cultivars [8]. Red- and purple-skinned cultivars (e.g., “Red Ruby”) contain high anthocyanin levels (3.23 mg/kg), which absorb blue–green light and competed with chlorophyll for excitation energy, resulting in 50% lower chlorophyll accumulation than that observed in white-skinned cultivars. Russet-skinned cultivars exhibit intermediate greening resistance because of the combined protective effect of suberin and pigments [8].

In this experiment, yellow-skinned cultivars (“Favorita”, “Lucinda”, “Jizhang shu 12”) were highly light-sensitive, showing severe color changes (e.g., “Favorita” blue light: ΔEab = 11.27) and high chlorophyll (18.47~18.52 mg/kg FW) and SGA (up to 290.41 mg/kg FW) accumulation. Red/purple-skinned cultivars (“Qing shu 9”, “Purple Potato”) delayed greening via anthocyanin-mediated light attenuation; their lower ΔEab (6.17~8.97) and chlorophyll (7.34~10.53 mg/kg FW) content confirm that anthocyanins compete with chlorophyll for light energy [8,39]. Notably, “Black Beauty” exhibited exceptionally high accumulation of anthocyanins (19.59 mg/kg FW) under white light. These results suggest that anthocyanins in “Qing shu 9” and “Purple Potato” act as light attenuators, absorbing high-energy blue–green photons, competing with chlorophyll, and restricting its accumulation in basal cells, which is consistent with the findings of Tanios’ and co-workers [39,48,49].

Genetic variation in SGA accumulation further differentiates greening resistance among cultivars. For example, under light treatment, the “Magnum Bonum” cultivar reached SGA levels of 680 mg/kg FW, exhibiting extraordinarily intense accumulation (approximately tenfold within 8 days), which was significantly higher than that of other cultivars (e.g., “Princess” increased by 6.7-fold and “Bintje” only 3.6-fold). This difference remained stable across a seven-year study, indicating that genetic factors play a dominant role [50]. Greening-prone cultivars such as “Gala” and “Juwel” accumulated SGA levels of 14.9–14.7 mg/100 g FW after 16 days of light treatment, approaching the toxicity threshold of 20 mg/100 g FW. By contrast, the “Albatros” cultivar exhibited no significant change in SGAs under light stress because of its high basal levels (12.4–17.1 mg/100 g FW), while the “Bavatop” cultivar even showed SGA degradation in the dark [51]. In this study, the α-solanine contents of “Jizhang shu 12” and “Lucinda” were 271.35 mg/kg FW and 131.99 mg/kg FW, respectively, while their α-chaconine contents reached 235.71 mg/kg FW and 290.41 mg/kg FW, which were significantly higher than those of “Long shu 10” (α-solanine: 45.9 mg/kg FW; α-chaconine: 152.24 mg/kg FW). Interestingly, “Long shu 10” exhibited significant tuber greening without a concomitant increase in SGAs, indicating that chlorophyll and SGA biosynthetic pathways are independently regulated [3]. Contrary to previous proposals that periderm thickness influences greening resistance, this study confirms that suberin biosynthesis is independent of chlorophyll and SGAs pathways [8]. Moreover, cultivars such as “White Rose” and “Yukon Gold” exhibited asynchronous changes in chlorophyll and SGAs accumulation, consistent with the established pattern [50].

The expression levels of *COP1* (1.4, 1.19) and *HY5* (1.33, 1.3) in “Long shu 10” and “Qing shu 9” were significantly higher than those in “Favorita” (0.06, 0.08), suggesting that enhanced light signal sensitivity may mediate greening differences among cultivars. The elevated expression of *COP1* and *HY5* in “Purple Potato” may be closely associated with its high anthocyanin content, suggesting that anthocyanins enhance cryptochrome-mediated *HY5* and *COP1* activity by modulating phytochrome signaling repression [4]. This observation is consistent with Sheng Xuan Liu’s conclusion that *HY5* participates in light signaling [4]. Notably, this study observed a positive correlation between *HY5* expression and SGA accumulation; for example, high *HY5* expression in “Purple Potato” was associated with low SGA levels. This contrasts with cultivars such as “Innovator” and “11FF35-2”, in which downregulated *HY5* expression corresponded to low SGAs accumulation. These findings suggest that the previously reported negative correlation between *HY5* and SGA accumulation contradicts the results of our study [4]. This discrepancy may result from tuber-specific regulation or anthocyanin-mediated competition for metabolic precursors.

“Favorita” mediated chlorophyll accumulation through *GUN4* (expression level: 0.09) (chlorophyll content: 18.47–18.52 mg/kg FW under blue light and white light), while “Jizhang shu 12” prioritized the upregulation of *CHLI1* (expression level: 0.45) (chlorophyll content: 13.09–16.50 mg/kg FW). Both cultivars are prone to greening. Although “Qing shu 9” showed high *CHLI1* expression (0.91), the light-attenuating effect of anthocyanins limited its chlorophyll content to 10.53–16.71 mg/kg FW. “Purple Potato” exhibited low expression of *CHID* (0.003), *GUN4* (0.12), and *CHLI1* (0.16), corresponding to low chlorophyll accumulation (7.34–11.35 mg/kg FW). Both cultivars are resistant to greening. Despite high *CHID* expression (0.256) and significant chlorophyll accumulation (12.37–17.26 mg/kg FW), the SGA content did not increase synchronously. This further confirms the independence of the chlorophyll synthesis pathway (dependent on *GUN4*, *CHID*, and *CHLI1*) and the SGA synthesis pathway [3]. Greening-susceptible cultivars (e.g., yellow-skinned ones) highly express 1–2 of these genes to promote chlorophyll synthesis, while greening-resistant cultivars (e.g., red/purple-skinned ones) restrict chlorophyll accumulation through low expression of these genes or anthocyanin masking.

Under light induction, the expression patterns of SGA biosynthetic genes varied among cultivars. For example, “Gala” showed significant SGA upregulation after one day of light exposure, whereas “Juwel” and “Krone” accumulated SGAs only after 7–16 days, indicating differences in transcriptional activation timing [51]. Furthermore, differences in light-induced responses may be associated with cholesterol and sugar substrate availability, as well as key enzyme activity (e.g., *SGT1*) [51]. Genes such as *HMGR1*, *SMO1-L*, and *TAM2* showed significantly greater induction amplitudes in “Magnum Bonum” compared with “Bintje” (e.g., a fivefold difference for *HMGR1*), whereas genes such as *SMT1* and *SQE* were more strongly repressed [50]. In this experiment, the expression level of the *SGT1* in “Favorita” (0.79) was 15-fold higher than in “Long shu 10” (0.05), directly contributing to differences in α-solanine and α-chaconine contents and confirming the key role of SGT1 as a solanine glycosyltransferase [15]. “Qing shu 9” and “Purple Potato” exhibited higher expression levels of SGA biosynthetic genes (*HMGR*, *SQS*, *SSR2*, and *SGT3*) but showed relatively low tuber SGA contents. This may be explained by metabolic pathway interactions, in which high anthocyanin biosynthesis competitively consumes phenylalanine precursors or reduces cholesterol conversion to SGAs by inhibiting the key sterol biosynthesis enzyme DWF1-L [52]. This finding is consistent with the conclusion by Nahar et al. that differences in SGA biosynthetic gene expression among cultivars lead to variations in tuber metabolic phenotypes. It further confirms that genotypic differences influence the isoprenoid and steroid metabolic branches of the SGA biosynthetic pathway by regulating the coordinated expression of key genes such as *HMGR*, *SQS*, *CAS1*, *SSR2*, *SGT1*, and *SGT3* [52].

Under white light, all cultivars showed a strong positive correlation (r = 0.6) among chlorophyll, carotenoids, anthocyanins, and SGAs, whereas in the dark, “Favorita” exhibited a negative correlation (r = 0.4) between chlorophyll and SGAs. At the genetic level, under UV-A light, the *CHID* and *HMGR* in “Favorita” exhibited a significant positive correlation (r = 0.72), whereas under blue light, the *CHLI1* and *SQS* in “Lucinda” showed a positive correlation (r = 0.65). These results indicate that light quality drives cultivar-specific metabolic phenotypes by enhancing the coordinated expression of chlorophyll and SGAs biosynthetic genes. These insights highlight key genetic targets for breeding greening-resistant cultivars and optimizing postharvest light management strategies to reduce food safety risks.

## 5. Conclusions and Perspectives

This study demonstrates that blue and white light exert the strongest effects on potato tuber greening and SGA biosynthesis, whereas green light has minimal impact. These effects are likely mediated by the light-signaling transcription factors HY5 and COP1, which regulate the expression of key genes associated with chlorophyll biosynthesis (*CHID*, *CHLI1*) and SGA biosynthesis (e.g., *HMGR*, *SGT3*). Yellow-skinned cultivars (“Favorita,” “Lucinda,” and “Jizhang shu 12”) are highly sensitive to blue and white light, with pronounced accumulation of chlorophyll and SGAs, whereas red/purple-skinned cultivars (“Qing shu 9,” “Purple Potato”) display delayed greening attributable to anthocyanin-mediated light attenuation. Notably, “Long shu 10” exhibits pronounced tuber greening but minimal SGA accumulation, confirming the independent regulation of these two metabolic pathways.

Based on these findings, future research can focus on two directions. First, from a technological perspective, lighting strategies in storage and retail environments should be optimized by reducing blue light exposure while increasing green light proportions, thereby mitigating light-induced greening and SGA accumulation. Second, from a breeding perspective, high-anthocyanin cultivars or varieties with independently regulated SGA pathways (e.g., “Long shu 10”) could be developed to resist light-induced greening, prolong shelf life, and reduce associated food safety risks. Molecular markers linked to key traits, such as anthocyanin biosynthesis genes and repressors of the SGA pathway, could facilitate the breeding of new cultivars with both low greening propensity and reduced SGA accumulation.

## Figures and Tables

**Figure 1 foods-14-03394-f001:**
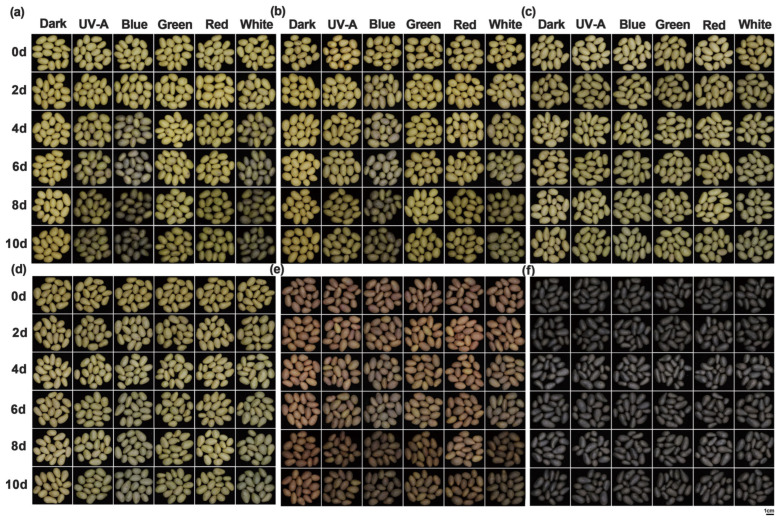
Greening phenotypes of tubers from six potato cultivars after 10-day light quality treatments. (**a**) “Favorita”; (**b**) “Lucinda”; (**c**) “Jizhang shu 12”; (**d**) “Long shu 10”; (**e**) “Qing shu 9”; (**f**) “Purple Potato”.

**Figure 2 foods-14-03394-f002:**
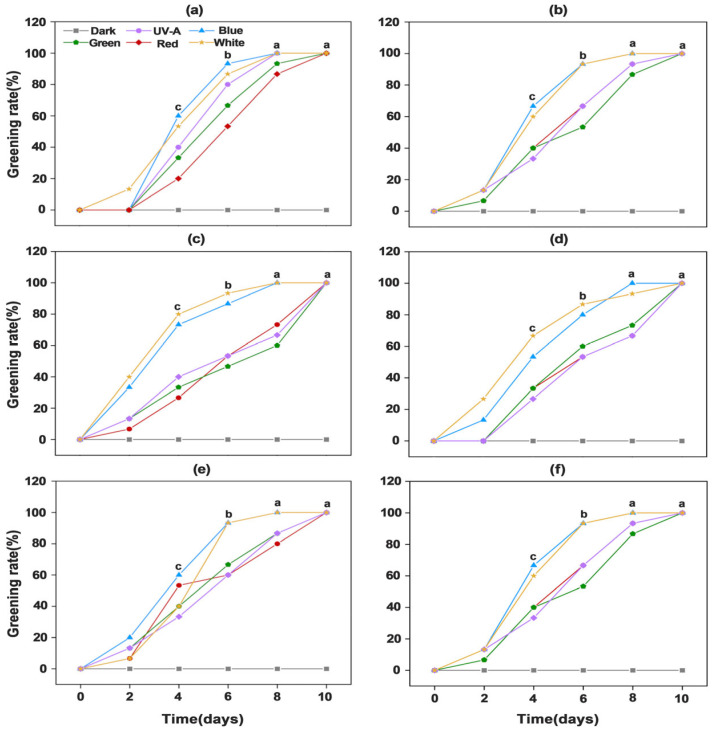
Analysis of 10-day greening rates in tubers of six potato cultivars under different light quality treatments. (**a**) “Favorita”; (**b**) “Lucinda”; (**c**) “Jizhang shu 12”; (**d**) “Long shu 10”; (**e**) “Qing shu 9”; (**f**) “Purple Potato”. Different letters indicate statistically significant differences among groups, with *p* < 0.05.

**Figure 3 foods-14-03394-f003:**
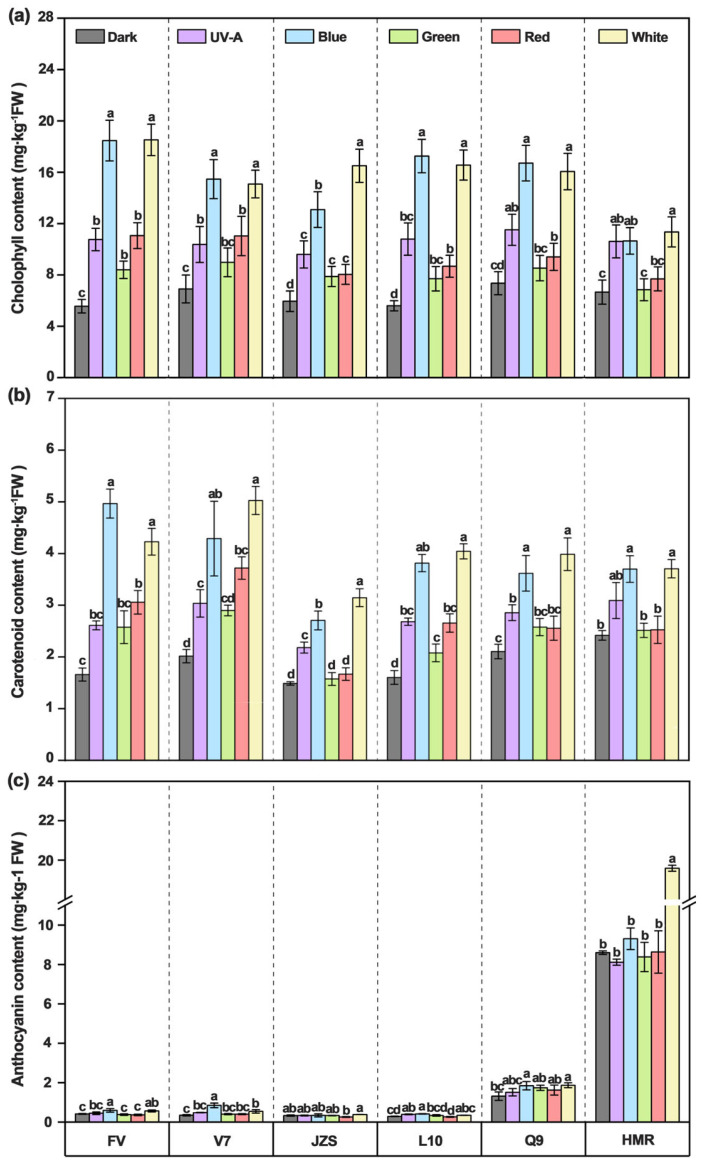
Pigment contents (chlorophyll, carotenoid, and anthocyanin) in tubers of six potato cultivars under 10-day light quality treatments. (**a**): chlorophyll content; (**b**) carotenoid content; (**c**) anthocyanin content. FV, Favorita; V7, Luxinda; JZS, Jizhang shu 12; L10, Long shu 10; Q9, Qing shu NO. 9; HMR: Purple Potato. Different letters indicate statistically significant differences among groups, with *p* < 0.05.

**Figure 4 foods-14-03394-f004:**
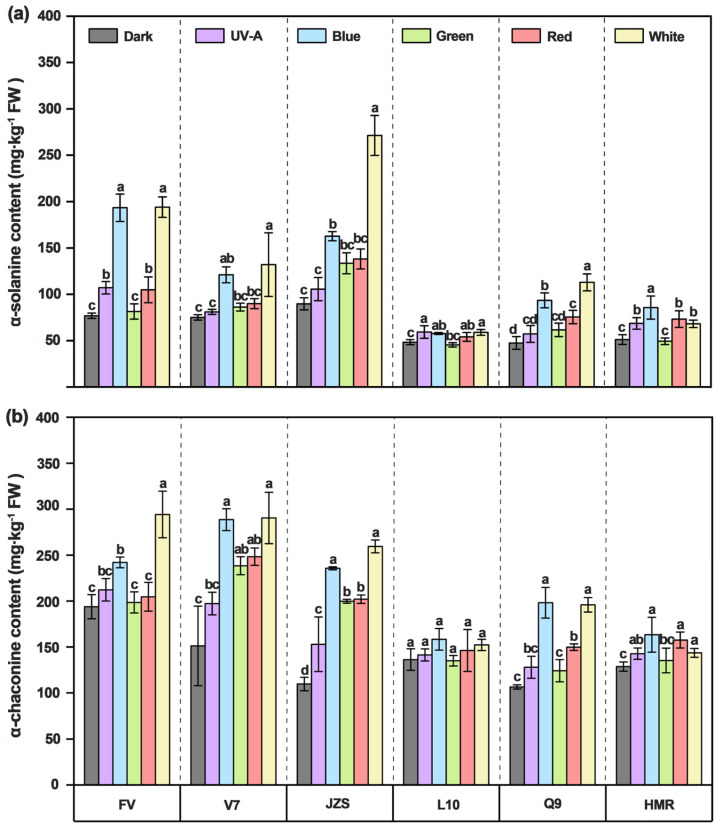
Changes in α-solanine and α-chaconine contents in tubers of six potato cultivars under 10 -day light quality treatments. (**a**) α-solanine content; (**b**) α-chaconine content. FV, Favorita; V7, Luxinda; JZS, Jizhang shu 12; L10, Long shu 10; Q9, Qing shu NO. 9; HMR: Purple Potato. Different letters indicate statistically significant differences among groups, with *p* < 0.05.

**Figure 5 foods-14-03394-f005:**
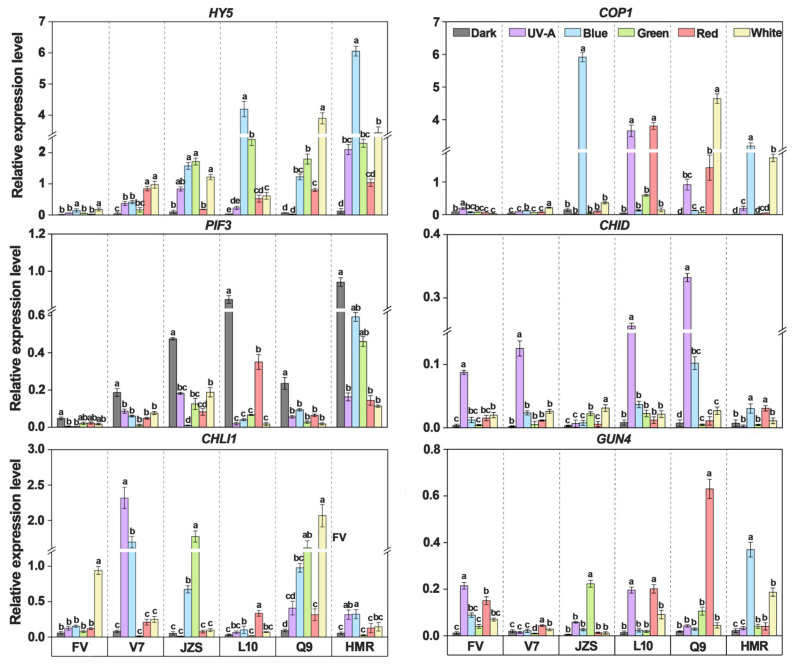
Expression changes of light signaling transcription factors and key genes involved in chlorophyll biosynthesis in tuber peels under different light quality treatments. FV, Favorita; V7, Luxinda; JZS, Jizhang shu 12; L10, Long shu 10; Q9, Qing shu NO. 9; HMR: Purple Potato. Different letters indicate statistically significant differences among groups, with *p* < 0.05.

**Figure 6 foods-14-03394-f006:**
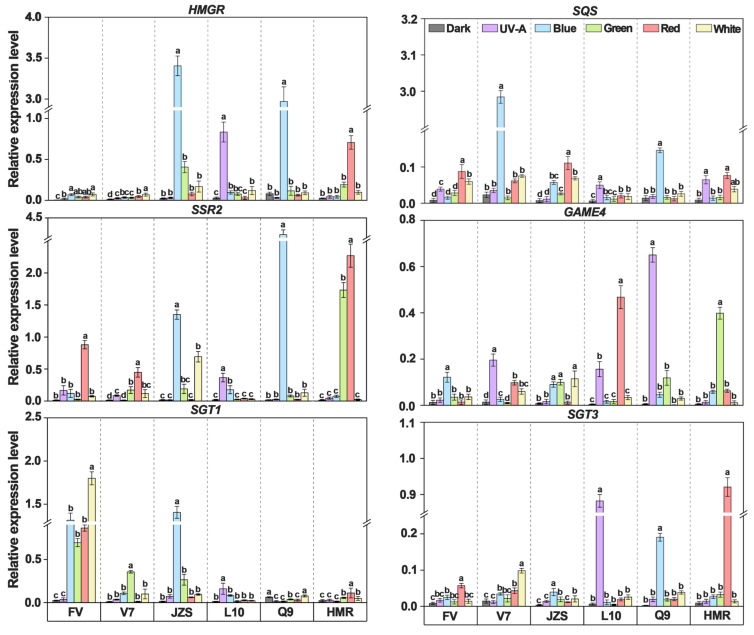
Expression changes of key genes involved in SGA biosynthesis in tuber peels under different light quality treatments. FV, Favorita; V7, Luxinda; JZS, Jizhang shu 12; L10, Long shu 10; Q9, Qing shu NO. 9, HMR: Purple Potato. Different letters indicate statistically significant differences among groups, with *p* < 0.05.

**Figure 7 foods-14-03394-f007:**
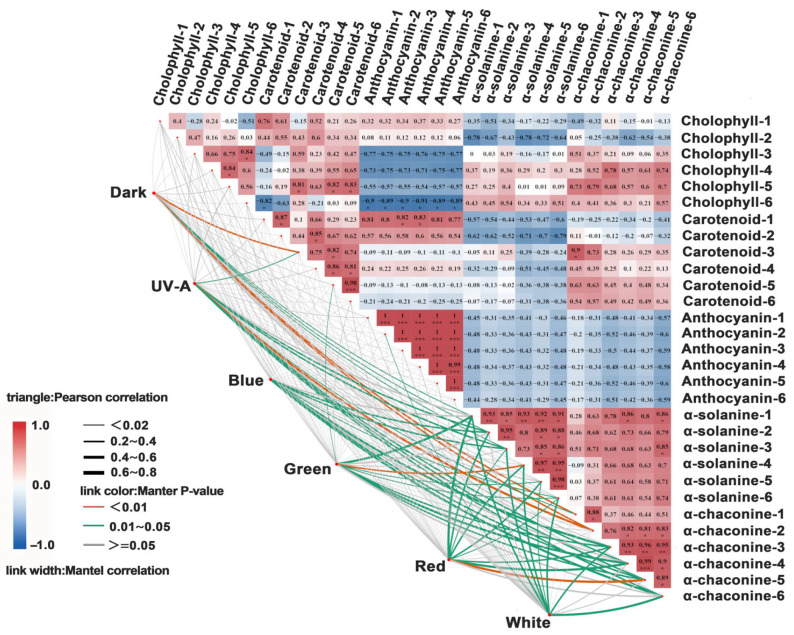
Correlation analysis of pigment and SGA contents in tuber peels under different light quality treatments. Indices 1–6 correspond to six potato cultivars: 1, Favorita; 2, Lucinda; 3, Jizhang shu 12; 4, Long shu 10; 5, Qing shu 9; 6, Purple Potato. Asterisks indicate statistical differences between treatments and each index, as determined by *t*-test (*p* < 0.05 *, *p* < 0.001 **, *p* < 0.0001 ***).

**Figure 8 foods-14-03394-f008:**
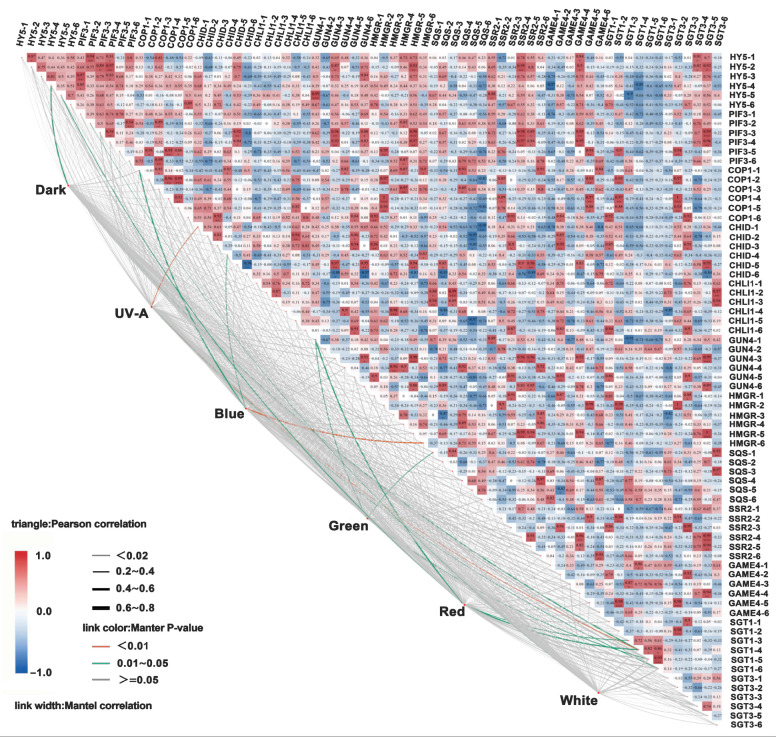
Correlation analysis of expression levels of key genes involved in chlorophyll and SGA biosynthesis in tuber peels under different light-quality treatments. Indices 1–6 correspond to six potato cultivars: 1, Favorita; 2, Lucinda; 3, Jizhang shu 12; 4, Long shu 10; 5, Qing shu 9; and 6, Purple Potato. Asterisks indicate statistical differences between treatments and each index, as determined by *t*-test (*p* < 0.05 *, *p* < 0.001 **, *p* < 0.0001 ***).

**Table 1 foods-14-03394-t001:** Color difference analysis of tubers from six potato cultivars following 10-day light quality treatments.

Cultivar	Light Quality	Δa*	ΔL*	Δb*	ΔE* ab
Favorita	Dark	1.35 a	2.88 a	11.50 a	11.93 a
	UV-A light	−3.33 c	−2.92 c	5.73 c	7.24 e
	Blue light	−5.30 d	−9.16 e	3.88 d	11.27 b
	Green light	−3.52 c	−2.25 c	7.72 b	8.78 c
	Red light	−2.20 b	−1.12 b	7.31 b	7.92 d
	White light	−5.14 d	−5.42 d	1.96 e	7.72 d
Lucinda	Dark	1.35 a	2.88 a	11.50 a	11.93 a
	UV-A light	−3.33 c	−2.92 c	5.73 c	8.24 c
	Blue light	−5.30 d	−9.16 e	3.88 d	9.27 b
	Green light	−3.52 c	−2.25 b	7.72 b	8.78 b
	Red light	−2.20 b	−2.12 b	7.31 b	7.92 d
	White light	−5.14 d	−5.42 d	1.96 e	7.72 d
Jizhang shu 12	Dark	1.35 a	2.88 a	11.50 a	11.13 a
	UV-A light	−3.33 c	−2.92 c	5.73 c	7.24 e
	Blue light	−5.30 d	−9.16 e	3.88 d	8.27 c
	Green light	−3.52 c	−2.25 b	7.72 b	8.78 b
	Red light	−2.20 b	−2.12 b	7.31 b	7.92 d
	White light	−5.14 d	−6.42 d	1.96 e	7.72 d
Long shu 10	Dark	1.35 a	2.28 a	11.50 a	10.93 a
	UV-A light	−3.33 c	−2.92 c	5.73 d	7.24 e
	Blue light	−5.30 d	−9.16 e	3.28 e	9.27 b
	Green light	−3.52 c	−2.25 b	6.72 c	8.78 c
	Red light	−2.20 b	−2.12 b	7.31 b	7.92 d
	White light	−5.14 d	−6.90 d	1.96 f	7.72 d
Qing shu 9	Dark	1.35 a	1.28 a	11.50 a	11.93 a
	UV-A light	−3.33 c	−2.92 c	5.73 d	9.04 b
	Blue light	−5.30 e	−9.16 e	3.20 e	6.17 e
	Green light	−3.92 d	−2.25 b	6.12 c	8.18 c
	Red light	−2.20 b	−2.12 b	6.81 b	7.92 d
	White light	−5.14 e	−7.72 d	1.96 f	5.72 e
Purple Potato	Dark	1.35 a	2.08 a	11.50 a	11.53 a
	UV-A light	−3.33 c	−2.92 c	5.73 d	7.24 e
	Blue light	−5.30 d	−9.16 e	3.88 e	8.97 b
	Green light	−3.52 c	−2.25 b	7.02 c	8.78 c
	Red light	−2.20 b	−2.12 b	9.31 b	7.92 d
	White light	−5.14 d	−7.42 d	1.96 f	7.72 d

Different letters indicate statistically significant differences among groups, with *p* < 0.05.

## Data Availability

The original contributions presented in this study are included in the article/Appendix A. Further inquiries can be directed to the corresponding authors.

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
