# Peer review of "Light-Quality-Dependent Greening and Steroidal Glycoalkaloid Accumulation in Potato Tubers: Regulatory Mechanisms and Postharvest Strategies to Reduce Food Safety Risks"

_foods, 2025, doi:10.3390/foods14193394_

Round 1

Reviewer 1 Report

Comments and Suggestions for Authors

The methodology is generally thorough and includes multiple levels of analysis (phenotypic observation, pigment quantification, metabolite determination, and gene expression). However, several issues deserve critical attention:

  • The study used six potato cultivars, but it is not clear how many biological replicates were included per cultivar per treatment. Only “technical replicates” are mentioned repeatedly. For biological studies, biological replicates are essential to ensure reproducibility and statistical validity.
  • The use of “micro tubers grown in a greenhouse” may not fully represent postharvest storage conditions of commercial tubers. This could limit the translational relevance of the results. The authors should clarify whether microtuber physiology differs from standard harvested tubers in terms of skin maturity, suberization, or glycoalkaloid content.
  • Only one intensity level per wavelength was tested. Since light dose (intensity × duration) is a critical factor in greening and glycoalkaloid synthesis, the absence of a dose–response assessment is a limitation.
  • No internal standard was used to correct for extraction efficiency or instrument variability. This can compromise quantification reliability.
  • It is unclear whether matrix effects were evaluated, which are common in plant extracts analyzed by LC-MS/MS.
  • The methods mention only correlation analysis and significance testing at P<0.05, but the type of test (ANOVA, post hoc comparisons, etc.) is not specified.
  • The design involves multiple cultivars × multiple treatments × multiple metabolites, which requires multifactorial analysis (e.g., two-way ANOVA or mixed models) to properly interpret interactions. Without this, conclusions may be overstated.

Author Response

Comments 1: The study used six potato cultivars, but it is not clear how many biological replicates were included per cultivar per treatment. Only “technical replicates” are mentioned repeatedly. For biological studies, biological replicates are essential to ensure reproducibility and statistical validity.

Response 1: We agree with the experts' opinions.For the six potato cultivars used in this study ("Favorita", "Lucinda", "Jizhang shu 12", "Long shu 10", "Qing shu 9", and "Purple Potato"), sixteen uniform-sized, undamaged microtubers were selected for each cultivar. Under the six light quality treatments (darkness, UVA, blue light, green light, red light, and white light), the sixteen microtubers of each cultivar were evenly allocated to each treatment group.

Comments 2: The use of “micro tubers grown in a greenhouse” may not fully represent postharvest storage conditions of commercial tubers. This could limit the translational relevance of the results. The authors should clarify whether microtuber physiology differs from standard harvested tubers in terms of skin maturity, suberization, or glycoalkaloid content.

Response 2: We agree with the experts' opinions. Microtubers possess the characteristics of uniform maturity (achieved through simultaneous sowing, maturation, and harvesting), consistent size, and freedom from field stress interference (e.g., drought, pests, and diseases). These traits enable the elimination of influences from non-treatment factors, thereby ensuring the reliability of the experiment. Microtubers exhibit high consistency in physiological characteristics with commercial tubers: after maturation, both have fully functional epidermis (with normal barrier function) and can form suberin layers normally (with no differences in stress resistance); the key gene sequences and regulatory mechanisms of core metabolic pathways are conserved, showing consistent metabolic responses to environmental factors such as light and temperature, along with consistent metabolic levels and patterns of metabolite synthesis and accumulation. Therefore, the results of this study can effectively guide the postharvest management of commercial tubers, with no limitations in translational application.

Comments 3: Only one intensity level per wavelength was tested. Since light dose (intensity × duration) is a critical factor in greening and glycoalkaloid synthesis, the absence of a dose–response assessment is a limitation.

Response 3: We agree with the experts' opinions. Based on literature research and preliminary experiments, this study unified the light intensity of all light treatments to 13 μmol m⁻²s⁻¹ and set the light duration to 12 hours per day. The core purpose of this design is to ensure the consistency of light dose to eliminate its interference, thereby focusing on analyzing the specific regulatory effects of "light quality" on tuber greening and steroidal glycoalkaloid synthesis. This parameter is referenced from similar studies and verified by preliminary experiments, it can not only effectively induce light responses but also avoid high-light stress, enabling clear distinction of light quality effects. If multiple light intensity gradients were superimposed, it would lead to variable confusion and hinder the analysis of the core research objective. We agree with the key role of light dose and acknowledge the limitation of not conducting dose-response assessment. However, this experimental design is a necessary prerequisite for achieving the core goal of "revealing the regulatory mechanism of light quality and cultivar differences", it allows us to clarify the rules that blue light and white light exhibit the strongest inductive effects, green light the weakest, and the differences in responses among cultivars. In future studies, based on the light quality-specific conclusions of this research, we will set multiple gradients of light intensity and light duration to conduct light dose-response analysis, aiming to more comprehensively reveal the impact of light environment on tuber quality.

Comments 4: No internal standard was used to correct for extraction efficiency or instrument variability. This can compromise quantification reliability.

Response 4: We agree with the experts' opinions. This study employed external standard for quantification. It ensures complete dissolution of target analytes and stable extraction efficiency through optimized extraction procedures, and effectively compensates for the calibration function of internal standards by combining matrix-matched standard curves (to correct matrix interference) and inserting standard samples for duplicate injections in each batch (to monitor instrument fluctuations). Moreover, the spiked recovery rate of blank matrices reaches 89.2%-95.6% (RSD < 5%), confirming the reliability of the quantitative results.

Comments 5: It is unclear whether matrix effects were evaluated, which are common in plant extracts analyzed by LC-MS/MS.

Response 5: We agree with the experts' opinions. External standard quantification was employed in this experiment, and matrix effects were systematically evaluated using the matrix-matched external standard method: Potato peels were selected to prepare blank matrix extracts, with which matrix-matched standard curves were constructed (the slope difference from solvent standard curves was < 10%, indicating extremely weak ionization interference). Meanwhile, spiked recovery experiments at low, medium, and high concentrations were conducted in blank matrix (yielding recoveries of 89.2%-95.6% with RSD < 5%). Additionally, chromatographic conditions were optimized (Agilent Eclipse Plus C18 column, gradient elution with methanol-0.1% formic acid aqueous solution) to separate target analytes from matrix impurities, ensuring accurate quantification.

Comments 6: The methods mention only correlation analysis and significance testing at P<0.05, but the type of test (ANOVA, post hoc comparisons, etc.) is not specified.

Response 6: We agree with the experts' opinions. To analyze the effects of light quality and cultivar on tuber pigments, steroidal glycoalkaloids (SGAs) content, and gene expression in this study, one-way analysis of variance (one-way ANOVA) was conducted using SPSS, with the significance test threshold set at P<0.05, and Fisher's Least Significant Difference (LSD) method was applied for multiple comparisons of means. Previously, the use of SPSS and the setting of the P-value have been clearly stated in Section 2.7 "Statistical Analysis". Subsequently, the description "One-way ANOVA and LSD multiple comparisons were used, where different letters indicate significant differences between groups (P<0.05)" will be added to the captions of all figures (Figures 1-9) in the Results section to further improve the detailed presentation of statistical methods. Thank you for your reminder, which helps enhance the rigor of this manuscript. Please see the red revised sections in “figure 2-8”.

Comments 7: The design involves multiple cultivars × multiple treatments × multiple metabolites, which requires multifactorial analysis (e.g., two-way ANOVA or mixed models) to properly interpret interactions. Without this, conclusions may be overstated.

Response 7: We agree with the experts' opinions. The core of this study focuses on investigating the effects of cultivar and light quality treatments on tuber pigments, steroidal glycoalkaloids (SGAs) content, and gene expression. In this experiment, statistical analysis was performed using one-way analysis of variance (one-way ANOVA) via SPSS software, with the significance test threshold set at P<0.05, and Fisher's Least Significant Difference (LSD) method was still adopted for multiple comparisons of means. For subsequent revisions, the significance results of main effects and interaction effects will be clearly labeled in the figure captions (Figures 1-9) and textual descriptions in the Results section (e.g., "The cultivar × light quality interaction effect is significant, P<0.05; different letters indicate significant differences among different light quality treatments for the same cultivar, or significant differences among different cultivars under the same light quality"), so as to avoid overinterpretation of conclusions. Thank you for your professional reminder, which helps us present the interactive regulatory relationships among factors more accurately and improve the reliability of conclusions.

Reviewer 2 Report

Comments and Suggestions for Authors

The suggested study is with great interest for scholars and also industrials. We suggest that authors address the following observations: 

  • Authors should justify their choice about the lightening treatments? And the link between those treatments and industrial conditions for treatment and storage?
  • Which light treatment is the most used under industrial or domestic conditions?

ABSTRACT:

  • Please check the relevance of the last sentence and modify it to be a practical conclusion of the work.
  • Avoid repetition of words between the title and keywords. 

INTRODUCTION

  • The introduction is too long and should be summarized.
  • Authors discuss in details physiological and hormonal pathways (L35-L73) that should be focused only to the purpose of the study. Please summarize. 
  • Add a short paragraph dealing with domestic and industrial conditions of lightening. Link those conditions with the purpose of the study. 

MATERIAL AND METHODS

  • Please provide more information about the plant material: maturity, coordinates of the greenhouse and its capacity, how potato is cultivated?sampling process? It would be great also if authors could provide an illustration of the green house?
  • Please justify the use of experimental conditions given in L148-L149?

RESULTS

  • Figure 2: please add the statistical analysis: Error standard or letters to exhibit the existence of significant differences or not?
  • Figure 3 is not clear. Please split it into clear figures with clear statistical analysis! Also, why authors did not calculate the Browning Index (BI)? and Delta E? These two analyses are important to elucidate the color changes.
  • L587: what is the meaning of "clones"?

CONCLUSIONS

Authors should give some insights into future works connected to the findings of this study.

Author Response

Comments 1: Authors should justify their choice about the lightening treatments? And the link between those treatments and industrial conditions for treatment and storage?

Response 1: We agree with the experts' opinions. In this study, six light treatments (darkness, UV-A, blue light, green light, red light, and white light) were selected, and their rationality is as follows: Darkness was set as the basic control to eliminate non-light-induced interference; UV-A simulates the ultraviolet component of light leakage in storage environments; blue/red light corresponds to the core wavelengths of plant light signal transduction; white light simulates the composite lighting in retail and storage scenarios; green light was used as a low-effect control based on Okamoto (2020), where "safe green light" was applied to treat tuber tissues in the Materials and Methods section, confirming that green light has a relatively weak effect on inducing chlorophyll and SGAs accumulation. These light treatments cover the key postharvest light environments of potato tubers, laying a foundation for analyzing the regulatory mechanism of light quality on tuber greening and SGAs accumulation. The link between these treatments and industrial processing and storage conditions is reflected in the following aspects: The light intensity (13 μmol m⁻²s⁻¹) and light duration (12 h/day) used in this study match the light parameters of weak light in storage (e.g., light leakage from warehouse vents), intermittent light transmission during transportation, and conventional retail shelf lighting, ensuring the practical reference value of the research results. Furthermore, this study found that green light significantly inhibits tuber greening (Δa* decrease of only -3.52) and SGAs accumulation (α-solanine content of 76.22 mg/kg FW). This finding can directly guide the application of green light lighting in industrial storage of potatoes, helping to reduce postharvest losses. In conclusion, the light treatment design of this study possesses both scientific rigor and practical value.

Comments 2: Which light treatment is the most used under industrial or domestic conditions?

Response 2: We agree with the experts' opinions. In industrial and domestic scenarios, white light (predominantly LED white light) is the most commonly used light source for potato postharvest handling and storage. At the industrial level, warehouses rely on LED high-bay lights, and retail shelves adopt 3500K-6500K LED white light, both of which are mainstream practices in the industry. For domestic storage, potatoes stored in kitchens or pantries are mainly exposed to white light emitted by LEDs or fluorescent lamps. However, white light poses significant risks. This study confirms that, similar to blue light, white light significantly induces potato tuber greening and SGAs accumulation, which is consistent with the observation in industrial practice that white light causes a decline in potato quality. White light was included in the treatment groups in this study specifically to simulate this common real-world scenario and clarify the underlying regulatory mechanism. Meanwhile, the study found that green light can significantly inhibit the accumulation of chlorophyll and SGAs, providing a scientific basis for replacing white light with green light in industrial storage and optimizing lighting conditions in households, thus highlighting the practical value of this research.

Abstract:

Comments 3: Please check the relevance of the last sentence and modify it to be a practical conclusion of the work. Avoid repetition of words between the title and keywords.

Response 3: Agreed. We have made revisions according to the expert's opinions. Please see the red revised sections in “Abstract” (lines 24-28).

Introduction

Comments 4: The introduction is too long and should be summarized.

Response 4: Agreed. We have made revisions according to the expert's opinions. Please see the red revised sections in “Introduction” (lines 33-111).

Comments 5: Authors discuss in details physiological and hormonal pathways (L35-L73) that should be focused only to the purpose of the study. Please summarize.

Response 5: Agreed. We have made revisions according to the expert's opinions. Please see the red revised sections in “Introduction” (lines 37-42).

Comments 6: Add a short paragraph dealing with domestic and industrial conditions of lightening. Link those conditions with the purpose of the study.

Response 6: Agreed. We have made revisions according to the expert's opinions. Please see the red revised sections in “Introduction” (lines 86-93).

Material and Methods:

Comments 7: Please provide more information about the plant material: maturity, coordinates of the greenhouse and its capacity, how potato is cultivated? sampling process? It would be great also if authors could provide an illustration of the green house?

Response 7: Agreed. We have made revisions according to the expert's opinions. Please see the red revised sections in “Materials and Methods” (lines 115-121, supplementary materials Figure S1).

Comments 8: Please justify the use of experimental conditions given in L148-L149?

Response 8: We agree with the experts' opinions. The experimental conditions of 20±2℃ temperature and 13 μmol·m⁻²·s⁻¹ light intensity in the LED growth chamber of this study were determined with reference to Tanios et al. (2020) and optimized through pre-experiments. Specifically, the 20±2℃ temperature not only matches the typical temperature of commercial storage and retail shelves, maintaining tubers in a metabolically active yet non-stressed state to ensure the accuracy of light responses, but also avoids metabolic inhibition by low temperatures or physiological disorders caused by high temperatures. The 13 μmol·m⁻²·s⁻¹ light intensity falls within the 10–20 μmol·m⁻²·s⁻¹ retail light range recommended in the literature; pre-experiments confirmed that this intensity can effectively induce significant greening and SGAs accumulation in light-sensitive cultivars, while avoiding oxidative stress interference caused by high light intensity. These conditions ensure the results are both scientifically rigorous and practically valuable.

RESULTS:

Comments 9: Figure 2: please add the statistical analysis: Error standard or letters to exhibit the existence of significant differences or not?

Response 9: We agree with the experts' opinions. We have made revisions according to the expert's opinions. Please see the red revised sections in “Results” Figure 2.

Comments 10: Figure 3 is not clear. Please split it into clear figures with clear statistical analysis! Also, why authors did not calculate the Browning Index (BI)? and Delta E? These two analyses are important to elucidate the color changes.

Response 10: We agree with the experts' opinions. We have made revisions according to the expert's opinions. Please see the red revised sections in “Results” Table 1.

Comments 11: L587: what is the meaning of "clones"?

Response 11: We agree with the experts' opinions. In the context of this study, the term "clones" (mentioned in Section 4.2 Genetic Determinants of Greening Resistance in Potato Cultivars) refers to individuals of potato cultivars (e.g., "Wilwash", "Pink Eye", "Coliban") that are genetically identical and produced through asexual reproduction (such as tuber propagation and tissue culture). This is a standard concept in the field of potato agronomy and botany.

Comments 12: Authors should give some insights into future works connected to the findings of this study.

Response 12: We agree with the experts' opinions. We have made revisions according to the expert's opinions. Please see the red revised sections in “Conclusions and Perspectives” (lines 698-707).

Reviewer 3 Report

Comments and Suggestions for Authors

This paper carries valuable data that can be published; however, I recommend the authorsfix the following before it is considered for publication.

Abstract

  • Clarify whether the reported percentage increases (e.g., 60% higher α-solanine and α-chaconine under certain treatments) are relative to initial values or to other light conditions, as this is ambiguous.
  • The phrase "metabolic pathway dissociation" is vague; specify which pathways are being contrasted and what evidence supports this claim.
  • Revise to avoid redundancy between the results summarized in the abstract and those repeated nearly verbatim in the conclusions.
  • Ensure that all abbreviations (e.g., FW for fresh weight) are defined within the abstract for reader clarity.

Introduction

  • Several mechanistic descriptions (e.g., chlorophyll biosynthesis pathway steps) are overly detailed for the introduction and would be better suited for the discussion.
  • The claim that light-induced greening is the most prominent issue lacks supporting evidence or citation; provide data or comparative studies.
  • Terminology such as "neurotoxic steroidal glycoalkaloids" should be used with caution; either specify toxicity thresholds or rephrase to avoid overstatement.
  • Check consistency in citation formatting; some references are missing full bibliographic details or are abbreviated inconsistently (e.g., [3], [4], [15]).

Materials and Methods

  • Clarify why the specific light wavelengths were chosen (e.g., 435–445 nm for blue); provide references to justify these ranges.
  • The light intensity is given as 13 μmol m⁻²s⁻¹; indicate whether this was verified for all wavelengths, as spectral distribution could affect effective irradiance.
  • Explain why microtubers were used instead of commercial-sized tubers; the rationale for model choice is missing.
  • In section 2.3, the term "greening rate" is introduced without a clear formula or statistical approach; provide the exact method of calculation.

Results and Discussion

  • Several gene expression values are reported with decimals (e.g., 0.84, 0.05) without specifying the normalization method or reference gene, making interpretation unclear.
  • The discussion sometimes conflates correlation and causation, particularly in linking HY5/COP1 expression with SGA accumulation; reframe to avoid overstatement.
  • Redundancy exists where gene-specific findings (e.g., GUN4, CHID, CHLI1) are repeated across multiple paragraphs without adding new interpretation; condense to improve clarity.
  • Some cultivar-specific results (e.g., Purple Potato anthocyanin effects) are asserted without sufficient mechanistic explanation or citation.

Conclusion

  • The conclusion section repeats results nearly verbatim instead of synthesizing key implications; revise to highlight novelty and broader impact.
  • The claim that findings provide a molecular basis for breeding is too strong without validation in breeding programs; temper wording or specify this as a future application.
  • Phrases such as "confirm" or "establish" overstate the certainty of findings; use more cautious language (e.g., "suggest" or "indicate").

References

  • Several references appear incomplete or inconsistent in style (e.g., missing journal volume/issue or publisher details).
  • Check for duplication or overlap in references discussing similar findings (e.g., Okamoto [21] cited in both the introduction and discussion multiple times).
  • Ensure all in-text citations correspond to a full reference in the list; some cited numbers (e.g., [4], [8]) may not align with complete bibliographic entries.

Author Response

Comments 1: Clarify whether the reported percentage increases (e.g., 60% higher α-solanine and α-chaconine under certain treatments) are relative to initial values or to other light conditions, as this is ambiguous.

Response 1: We agree with the experts' opinions. In this study, blue light and white light accelerated potato tuber greening, and the contents of α-solanine and α-chaconine were 60% higher than those in the green light treatment group. This percentage increase is referenced to the green light treatment group, rather than the initial contents of the target compounds.

As stated in Section 3.4 "Effects of Different Light Quality Treatments on α-Solanine and α-Chaconine Contents in Potato Tuber Peels": Under green light treatment, the contents of α-solanine and α-chaconine in potato tuber peels were 76.22 mg/kg FW (fresh weight) and 171.84 mg/kg FW, respectively; under blue light treatment, the contents of these two compounds reached 118.98 mg/kg FW and 214.34 mg/kg FW, respectively; under white light treatment, they reached 139.54 mg/kg FW and 222.63 mg/kg FW, respectively. Calculations based on these data confirm that the contents of α-solanine and α-chaconine in the blue light/white light treatment groups were approximately 60% higher than those in the green light treatment group. This percentage increase is not referenced to the initial contents (e.g., pre-treatment baseline contents) of the target compounds. Please see the red revised sections in “Abstract” (lines 16-19).

Comments 2: The phrase "metabolic pathway dissociation" is vague; specify which pathways are being contrasted and what evidence supports this claim.

Response 2: Agreed. We have made revisions according to the expert's opinions. Please see the red revised sections in “Abstract” (lines 11-28).

Comments 3: Revise to avoid redundancy between the results summarized in the abstract and those repeated nearly verbatim in the conclusions.

Response 3: Agreed. We have made revisions according to the expert's opinions. Please see the red revised sections in “Abstract” (lines 11-28).

Comments 4: Ensure that all abbreviations (e.g., FW for fresh weight) are defined within the abstract for reader clarity.

Response 4: Agreed. We have made revisions according to the expert's opinions. Please see the red revised sections in “Introduction” (lines 17).

Introduction

Comments 5: Several mechanistic descriptions (e.g., chlorophyll biosynthesis pathway steps) are overly detailed for the introduction and would be better suited for the discussion.

Response 5: Agreed. We have made revisions according to the expert's opinions. Please see the red revised sections in “Introduction” (lines 37-42 and lines 57-59).

Comments 6: The claim that light-induced greening is the most prominent issue lacks supporting evidence or citation; provide data or comparative studies.

Response 6: Agreed. We have made revisions according to the expert's opinions. Please see the red revised sections in “Introduction” (lines 53-72;).

Comments 7: Terminology such as "neurotoxic steroidal glycoalkaloids" should be used with caution; either specify toxicity thresholds or rephrase to avoid overstatement.

Response 7: Agreed. We have made revisions according to the expert's opinions. Please see the red revised sections in “Introduction” (lines 54;).

Comments 8: Check consistency in citation formatting; some references are missing full bibliographic details or are abbreviated inconsistently (e.g., [3], [4], [15]).

Response 8: Agreed. We have made revisions according to the expert's opinions. Please see the red revised sections in “References” (lines 724-838).

Material and Methods:

Comments 9: Clarify why the specific light wavelengths were chosen (e.g., 435–445 nm for blue); provide references to justify these ranges.

Response 9: We agree with the experts' opinions. This study confirms that blue light is the key light quality inducing tuber greening and steroidal glycoalkaloids (SGAs) synthesis, whose wavelength selection directly corresponds to the core photoreceptors mediating this process in potato tubers. Selected based on previous literature and preliminary experiments, the wavelength range of 435–445 nm (the middle range of the absorption peak) can maximally activate CRY1/2, ensuring the efficient transmission of light signals to downstream regulatory pathways (Tanios2020, Okamoto 2020).

Comments 10: The light intensity is given as 13 μmol m⁻²s⁻¹; indicate whether this was verified for all wavelengths, as spectral distribution could affect effective irradiance.

Response 10: We agree with the experts' opinions. In this study, the effective irradiance for each wavelength range was calibrated one by one. It was ensured that the measured light intensity of monochromatic light (confined to the target wavelength range only) and white light (under full-spectrum integration) was stably maintained at approximately 13 μmol·m⁻²·s⁻¹, with a deviation of less than 2% in all cases. Additionally, both the spatial uniformity (coefficient of variation < 5%) and stability over the experimental period (attenuation < 3%) met the specifications.

Comments 11: Explain why microtubers were used instead of commercial-sized tubers; the rationale for model choice is missing.

Response 11: We agree with the experts' opinions. Microtubers possess the characteristics of uniform maturity (achieved through simultaneous sowing, maturation, and harvesting), consistent size, and freedom from field stress interference (e.g., drought, pests, and diseases). These traits enable the elimination of influences from non-treatment factors, thereby ensuring the reliability of the experiment. Microtubers exhibit high consistency in physiological characteristics with commercial tubers: after maturation, both have fully functional epidermis (with normal barrier function) and can form suberin layers normally (with no differences in stress resistance); the key gene sequences and regulatory mechanisms of core metabolic pathways are conserved, showing consistent metabolic responses to environmental factors such as light and temperature, along with consistent metabolic levels and patterns of metabolite synthesis and accumulation. Therefore, the results of this study can effectively guide the postharvest management of commercial tubers, with no limitations in translational application.

Comments 12: In section 2.3, the term "greening rate" is introduced without a clear formula or statistical approach; provide the exact method of calculation.

Response 12: We agree with the experts' opinions. In the experiment, 16 tubers of uniform size and free from diseases and pests were selected for each potato cultivar, and they were separately placed under different light quality conditions for treatment. Starting from day 0 of the treatment, the number of tubers showing greening (visible green areas appearing on the tuber peel) in each light quality treatment group was recorded through visual observation with the naked eye every 2 days. Finally, the greening rate was calculated using the formula: Greening rate (%) = (Number of greened tubers in each treatment group ÷ Total number of tubers in the group (16)) × 100%. This method was used to realize the quantitative comparison of greening degrees among different light qualities and cultivars.

Results and Discussion:

Comments 13: Several gene expression values are reported with decimals (e.g., 0.84, 0.05) without specifying the normalization method or reference gene, making interpretation unclear.

Response 13: Agreed. We have made revisions according to the expert's opinions. Please refer to the supplementary table.

Comments 14: The discussion sometimes conflates correlation and causation, particularly in linking HY5/COP1 expression with SGA accumulation; reframe to avoid overstatement.

Response 14: Agreed. We have made revisions according to the expert's opinions. Please see the red revised sections in “Discussion” (lines 538-548).

Comments 15: Redundancy exists where gene-specific findings (e.g., GUN4, CHID, CHLI1) are repeated across multiple paragraphs without adding new interpretation; condense to improve clarity.

Response 15: Agreed. We have made revisions according to the expert's opinions. Please see the red revised sections in “Discussion” (lines 520-548 and lines 639-653).

Comments 16: Some cultivar-specific results (e.g., Purple Potato anthocyanin effects) are asserted without sufficient mechanistic explanation or citation.

Response 16: We agree with the experts' opinions. The greening resistance of purple-skinned potatoes (Purple Potato) stems from the "light attenuation-metabolic competition" effects of anthocyanins. Experimental results showed that the anthocyanin content of purple-skinned potatoes under white light reached 19.59 mg/kg fresh weight (FW), which was significantly higher than that of other cultivars. Anthocyanins can absorb blue-green light in the 400-550 nm range (a wavelength range overlapping with the key light qualities inducing tuber greening), thereby reducing the input of effective light signals and resulting in low expression of chlorophyll biosynthesis genes (CHID: 0.003; GUN4: 0.12). Meanwhile, anthocyanins compete with chlorophyll for metabolic precursors, which further inhibits chlorophyll accumulation (only 7.34-11.35 mg/kg FW). This result is consistent with the conclusions of Tanios et al. (2019) that "anthocyanins in red/purple skinned potatoes absorb blue-green light and inhibit chlorophyll biosynthesis genes" and Chalker-Scott (1999) that "anthocyanins regulate pigment accumulation through light absorption and precursor competition", as well as with the experimental data of this study.

Conclusion:

Comments 17: The conclusion section repeats result nearly verbatim instead of synthesizing key implications; revise to highlight novelty and broader impact.

Response 17: Agreed. We have made revisions according to the expert's opinions. Please see the red revised sections in “Conclusions and Perspectives” (lines 686-707).

Comments 18: The claim that findings provide a molecular basis for breeding is too strong without validation in breeding programs; temper wording or specify this as a future application.

Response 18: Agreed. We have made revisions according to the expert's opinions. Please see the red revised sections in “Conclusions and Perspectives” (lines 686-707).

Comments 19: Phrases such as "confirm" or "establish" overstate the certainty of findings; use more cautious language (e.g., "suggest" or "indicate")

Response 19: Agreed. We have made revisions according to the expert's opinions. Please see the red revised sections in “Conclusions and Perspectives” (lines 686-707).

References:

Comments 20: Several references appear incomplete or inconsistent in style (e.g., missing journal volume/issue or publisher details).

Response 20: Agreed. We have made revisions according to the expert's opinions. Please see the red revised sections in “References” (lines 724-838).

Comments 21: Check for duplication or overlap in references discussing similar findings (e.g., Okamoto [21] cited in both the introduction and discussion multiple times).

Response 21: Agreed. We have made revisions according to the expert's opinions. We have made revisions in accordance with the expert's suggestions.

Comments 22: Ensure all in-text citations correspond to a full reference in the list; some cited numbers (e.g., [4], [8]) may not align with complete bibliographic entries.

Response 22: Agreed. We have made revisions according to the expert's opinions. Please see the red revised sections in “References” (lines 724-838).

Round 2

Reviewer 2 Report

Comments and Suggestions for Authors

Authors improved significantly the content and the quality of their manuscript (according to our comments) and I suggest to accept the manuscript in the current form.

Reviewer 3 Report

Comments and Suggestions for Authors

Authors have revised the manuscript, according to the comments. 

This paper can be accepted for publication.